# ENHANCING CERTIFIED ROBUSTNESS OF SMOOTHED CLASSIFIERS VIA WEIGHTED MODEL ENSEMBLING

## ABSTRACT

Randomized smoothing has achieved state-of-the-art certified robustness against $l_2$-norm adversarial attacks. However, it is not wholly resolved on how to find the optimal base classifier for randomized smoothing. In this work, we employ a Smoothed WEighted ENsembling (SWEEN) scheme to improve the performance of randomized smoothed classifiers. We show the ensembling generality that SWEEN can help achieve optimal certified robustness. Furthermore, theoretical analysis proves that the optimal SWEEN model can be obtained from training under mild assumptions. We also develop an adaptive prediction algorithm to reduce the prediction and certification cost of SWEEN models. Extensive experiments show that SWEEN models outperform the upper envelope of their corresponding candidate models by a large margin. Moreover, SWEEN models constructed using a few small models can achieve comparable performance to a single large model with a notable reduction in training time.

## 1 INTRODUCTION

Deep neural networks have achieved great success in image classification tasks. However, they are vulnerable to *adversarial examples*, which are small imperceptible perturbations on the original inputs that can cause misclassification (Biggio et al., 2013; Szegedy et al., 2014). To tackle this problem, researchers have proposed various defense methods to train classifiers robust to adversarial perturbations. These defenses can be roughly categorized into empirical defenses and certified defenses. One of the most successful empirical defenses is adversarial training (Kurakin et al., 2017; Madry et al., 2018), which optimizes the model by minimizing the loss over adversarial examples generated during training. Empirical defenses produce models robust to certain attacks without a theoretical guarantee. Most of the empirical defenses are heuristic and subsequently broken by more sophisticated adversaries (Carlini & Wagner, 2017; Athalye et al., 2018; Uesato et al., 2018; Tramer et al., 2020). Certified defenses, either exact or conservative, are introduced to mitigate such deficiency in empirical defenses. In the context of $l_p$ norm-bounded perturbations, exact methods report whether an adversarial example exists within an $l_p$ ball with radius $r$ centered at a given input $x$. Exact methods are usually based on Satisfiability Modulo Theories (Katz et al., 2017; Ehlers, 2017) or mixed-integer linear programming (Lomuscio & Maganti, 2017; Fischetti & Jo, 2017), which are computationally inefficient and not scalable (Tjeng et al., 2019). Conservative methods are more computationally efficient, but might mistakenly flag a safe data point as vulnerable to adversarial examples (Raghunathan et al., 2018a; Wong & Kolter, 2018; Wong et al., 2018; Gehr et al., 2018; Mirman et al., 2018; Weng et al., 2018; Zhang et al., 2018; Raghunathan et al., 2018b; Dvijotham et al., 2018b; Singh et al., 2018; Wang et al., 2018b; Salman et al., 2019b; Croce et al., 2019; Gowal et al., 2018; Dvijotham et al., 2018a; Wang et al., 2018a). However, both types of defenses are not scalable to practical networks that perform well on modern machine learning problems (e.g., the ImageNet (Deng et al., 2009) classification task).

Recently, a new certified defense technique called *randomized smoothing* has been proposed (Lecuyer et al., 2019; Cohen et al., 2019). A (randomized) smoothed classifier is constructed from a base classifier, typically a deep neural network. It outputs the most probable class given by its base classifier under a random noise perturbation of the input. Randomized smoothing is scalable due to its independency over architectures and has achieved state-of-the-art certified $l_2$-robustness. In theory, randomized smoothing can apply to any classifiers. However, naively using randomized smoothing on standard-trained classifiers leads to poor robustness results. It is still not wholly

resolved on how a base classifier should be trained so that the corresponding smoothed classifier has good robustness properties. Recently, Salman et al. (2019a) employ adversarial training to train base classifiers and substantially improve the performance of randomized smoothing, which indicates that techniques originally proposed for empirical defenses can be useful in finding good base classifiers for randomized smoothing.

In this paper, we take a step towards finding suitable base models for randomized smoothing by model ensembling. The idea of model ensembling has been used in various empirical defenses against adversarial examples and shows promising results for robustness (Liu et al., 2018; Strauss et al., 2018; Pang et al., 2019; Wang et al., 2019; Meng et al., 2020; Sen et al., 2020). Moreover, an ensemble can combine the strengths of candidate models[1] to achieve superior clean accuracy (Hansen & Salamon, 1990; Krogh & Vedelsby, 1994). Thus, we believe ensembling several smoothed models can help improve both the robustness and accuracy. Specifically for randomized smoothing, the smoothing operator is commutative with the ensembling operator: ensembling several smoothed models is equivalent to smoothing an ensembled base model. This property makes the combination suitable and efficient. Therefore, we directly ensemble a base model by taking some pre-trained models as candidates and optimizing the optimal weights for randomized smoothing. We refer to the final model as a Smoothed WEighted ENsembling (SWEEN) model. Moreover, SWEEN does not limit how individual candidate classifiers are trained, thus is compatible with most previously proposed training algorithms on randomized smoothing.

Our contributions are summarized as follows:

1. We propose SWEEN to substantially improve the performance of smoothed models. Theoretical analysis shows the ensembling generality and the optimization guarantee: SWEEN can achieve optimal certified robustness w.r.t. the defined $\gamma$-robustness index, which is an extension of previously proposed criteria of certified robustness (Lemma 1), and SWEEN can be easily trained to a near-optimal risk with a surrogate loss (Theorem 2).

2. We develop an adaptive prediction algorithm for the weighted ensembling, which effectively reduces the prediction and certification cost of the smoothed ensemble classifier.

3. We evaluate our proposed method through extensive experiments. On all tasks, SWEEN models consistently outperform the upper envelopes of their respective candidate models in terms of the approximated certified accuracy by a large margin. In addition, SWEEN models can achieve comparable or superior performance to a large individual model using a few candidates with a notable reduction in total training time.

## 2  RELATED WORK

In the past few years, numerous defenses have been proposed to build classifiers robust to adversarial examples. Our work typically involves randomized smoothing and model ensembling.

**Randomized smoothing**  Randomized smoothing constructs a smoothed classifier from a base classifier via convolution between the input distribution and certain noise distribution. It is first proposed as a heuristic defense by (Liu et al., 2018; Cao & Gong, 2017). Lecuyer et al. (2019) first prove robustness guarantees for randomized smoothing utilizing tools from differential privacy. Subsequently, a stronger robustness guarantee is given by Li et al. (2018). Cohen et al. (2019) provide a tight robustness bound for isotropic Gaussian noise in $l_2$ robustness setting. The theoretical properties of randomized smoothing in various norm and noise distribution settings have been further discussed in the literature (Blum et al., 2020; Kumar et al., 2020; Yang et al., 2020; Lee et al., 2019; Teng et al., 2019; Zhang et al., 2020). Recently, a series of works (Salman et al., 2019a; Zhai et al., 2020) develop practical algorithms to train a base classifier for randomized smoothing. Our work improves the performance of smoothed classifiers via weighted ensembling of pre-trained base classifiers.

**Model ensembling**  Model ensembling has been widely studied and applied in machine learning as a technique to improve the generalization performance of the model (Hansen & Salamon, 1990; Krogh & Vedelsby, 1994). Krogh & Vedelsby (1994) show that ensembles constructed from accurate

---

[1]In this paper, "candidate model" and "candidate" refer to an individual model used in an ensemble. The term "base model" refers to a model to which randomized smoothing applies.

and diverse networks perform better. Recently, simple averaging of multiple neural networks has been a success in ILSVRC competitions (He et al., 2016; Krizhevsky et al., 2017; Simonyan & Zisserman, 2015). Model ensembling has also been used in defenses against adversarial examples (Liu et al., 2018; Strauss et al., 2018; Pang et al., 2019; Wang et al., 2019; Meng et al., 2020; Sen et al., 2020). Wang et al. (2019) have shown that a jointly trained ensemble of noise injected ResNets can improve clean and robust accuracies. Recently, Meng et al. (2020) find that ensembling diverse weak models can be quite robust to adversarial attacks. Unlike the above works, which are empirical or heuristic, we employ ensembling in randomized smoothing to provide a theoretical robustness certification.

## 3 PRELIMINARIES

**Notation**  Let $\mathcal{Y} = \{1, 2, ..., M\}$. We overload notation slightly, letting $k$ refer the $M$-dimensional one-hot vector whose $k$-th entry is 1 for $k = 1, ..., M$ as well. The choice should be clear from context. Let $\Delta_k = \{(p_1, p_2, ..., p_k) | p_i \geq 0, \sum_{i=1}^{k} p_i = 1\}$ be the $k$-dimensional probability simplex for $k \in \mathbb{N}_+$, and $\Delta = \Delta_M$. For an $M$-dimensional function $f$, we use $f_i$ to refer to its $i$-th entry, $i = 1, 2, \cdots, M$. We use $\mathcal{N}(0, \sigma^2 I)$ to denote the $d$-dimensional Gaussian distribution with mean 0 and variance $\sigma^2 I$. We use $\Phi^{-1}$ to denote the inverse of the standard Gaussian CDF, and use $\Gamma$ to denote the gamma function. We use $\mathbb{R}^*$ to denote the set of non-negative real numbers. For $x, a, b \in \mathbb{R}, a \leq b$, we define $\mathrm{clip}(x; a, b) = \min\{\max\{x, a\}, b\}$. We use $\Omega(\cdot)$ to denote Big-Omega notation that suppresses multiplicative constants.

**Neural network and classifier**  Consider a classification problem from $\mathcal{X} \subseteq \mathbb{R}^d$ to classes $\mathcal{Y}$. Assume the input space $\mathcal{X}$ has finite diameter $D = \sup_{x_1, x_2 \in \mathcal{X}} \|x_1 - x_2\|_2 < \infty$. The training set $\{(x_i, y_i)\}_{i=1}^{n}$ is *i.i.d.* drawn from the data distribution $\mathcal{D}$. We call $f$ a probability function or a classifier if it is a mapping from $\mathbb{R}^d$ to $\Delta$ or $\mathcal{Y}$, respectively. For a probability function $f$, its induced classifier $f^*$ is defined such that $f^*(x) = \arg\max_{1 \leq i \leq M} f_i(x)$. For simplicity, we will not distinguish between $f$ and $f^*$ when there is no ambiguity, and hence all definitions and properties for classifiers automatically apply to probability functions as well. $f(\cdot; \theta)$ denotes a neural network parameterized by $\theta \in \Theta$. Here $\Theta$ can include hyper-parameters, thus the architectures of $f(\cdot; \theta)$'s do not have to be identical.

**Certified robustness**  We call $x + \delta$ an adversarial example of a classifier $f$, if $f$ correctly classifies $x$ but $f(x + \delta) \neq f(x)$. Usually $\|\delta\|_2$ is small enough so $x + \delta$ and $x$ appear almost identical for the human eye. The ($l_2$-)robust radius of $f$ is defined as

$$r(x, y; f) = \inf_{F(x+\delta) \neq y} \|\delta\|_2, \tag{1}$$

which is the radius of the largest $l_2$ ball centered at $x$ within which $f$ consistently predicts the true label $y$ of $x$. Note that $r(x, y; f) = 0$ if $f(x) \neq y$. As mentioned before, we can extend the above definitions to the case when $f$ is a probability function by considering the induced classifier $f^*$. A certified robustness method tries to find some lower bound $r_c(x, y; f)$ of $r(x, y; f)$, and we call $r_c$ a certified radius of $f$.

**Randomized smoothing**  Let $f$ be a probability function or a classifier. The (randomized) smoothed function of $f$ is defined as

$$g(x) = \mathbb{E}_{\delta \sim \mathcal{N}(0, \sigma^2 I)}[f(x + \delta)]. \tag{2}$$

The (randomized) smoothed classifier of $f$ is then defined as $g^*$. Cohen et al. (2019) first provide a tight robustness guarantee for classifier-based smoothed classifiers, which is summerized in the following theorem:

**Theorem 1.** *(Cohen et al. (2019)) For any classifier $f$, denote its smoothed function by $g$. Then*

$$r(x, y; g) \geq \frac{\sigma}{2}[\Phi^{-1}(g_y(x)) - \Phi^{-1}(\max_{k \neq y} g_k(x))]. \tag{3}$$

Later on, Salman et al. (2019a); Zhai et al. (2020) extends Theorem 1 for probability functions.

## 4 SWEEN: SMOOTHED WEIGHTED ENSEMBLING

In this section, we describe the SWEEN framework we use. We also present some theoretical results for SWEEN models. The proofs of the results in this section can be found in Appendix A.

### 4.1 SWEEN: OVERVIEW

To be specific, we adopt a data-dependent weighted average of neural networks to serve as the base model for smoothing. Suppose we have some pre-trained neural networks $f(\cdot; \theta_1), ..., f(\cdot; \theta_K)$ as ensemble candidates. A weighted ensemble model is then

$$f_{ens}(\cdot; \theta, w) = \sum_{k=1}^{K} w_k f(\cdot; \theta_k), \tag{4}$$

where $\theta = (\theta_1, \cdots, \theta_K) \in \Theta^K$, and $w \in \Delta_K$ is the ensemble weight. For a specific $f_{ens}$, the corresponding SWEEN model is defined as the smoothed function of $f_{ens}$, denoted by $g_{ens}$. We have

$$g_{ens}(x; \theta, w) = \mathbb{E}_\delta[\sum_{k=1}^{K} w_k f(x + \delta; \theta_k)] = \sum_{k=1}^{K} w_k \mathbb{E}_\delta f(x + \delta; \theta_k) = \sum_{k=1}^{K} w_k g(x; \theta_k), \tag{5}$$

where $g(\cdot; \theta)$ is the smoothed function of $f(\cdot; \theta)$. This result means that $g_{ens}$ is the weighted sum of the smoothed functions of the candidate models under the same weight $w$, or more briefly, randomized smoothing and weighted ensembling are commutative. Thus, ensembling under the randomized smoothing framework can provides benefits in improving the accuracy and robustness.

To find the optimal SWEEN model, we can minimize a surrogate loss of $g_{ens}$ over the training set to obtain the value of appropriate weights. These data-dependent weights can make the ensemble model robust to the presence of some biased candidate models, as they will be assigned with small weights.

### 4.2 CERTIFIED ROBUSTNESS OF SWEEN MODELS

For a smoothed function $g$, the certified radius at $(x, y)$ provided by Theorem 1 is $r_c(x, y; g) = \text{clip}(\frac{\sigma}{2}[\Phi^{-1}(g_y(x)) - \Phi^{-1}(\max_{k \neq y} g_k(x))]; 0, D)$. We now formally define $\gamma$-robustness index as a criterion of certified robustness.

**Definition 1.** *($\gamma$-robustness index). For $\gamma : \mathbb{R}^* \to \mathbb{R}^*$ and a smoothed function $g$, the $\gamma$-robustness index of $g$ is defined as*

$$\mathcal{I}_\gamma(g) = \mathbb{E}_{(x,y)\sim\mathcal{D}}\gamma(r_c(x, y; g)). \tag{6}$$

It can be easily observed that $\gamma$-robustness index is an extension of many frequently-used criteria of certified robustness of smoothed classifiers.

**Proposition 1.** *Let $\gamma_1(r) = \mathbb{1}\{r \geq R\}, \gamma_2(r) = r, \gamma_3(r) = \frac{\pi^{\frac{d}{2}}}{\Gamma(\frac{d}{2}+1)} r^d$. Then, $\gamma_1$-robustness index is the certified accuracy at radius $R$ (Cohen et al., 2019); $\gamma_2$-robustness index is the average certified radius (Zhai et al., 2020); $\gamma_3$-robustness index is the average volume of the certified region.*

We note that criteria considering the volume of the certified region are more comprehensive than those only considering the certified radii in a sense, as they take the input dimension into account.

Now consider $\mathcal{F} = \{f(\cdot; \theta) : \mathbb{R}^d \to \Delta | \theta \in \Theta\}$, the set of neural networks parametrized over $\Theta$. The corresponding set of smoothed functions is $\mathcal{G} = \{g(x; \theta) = \mathbb{E}_{\delta\sim\mathcal{N}(0,\sigma^2 I)}[f(x + \delta; \theta)] | \theta \in \Theta\}$. Suppose $\theta_1, \cdots, \theta_K$ are drawn *i.i.d.* from a fixed probability distribution $p$ on $\Theta$. The set of SWEEN models is then

$$\hat{\mathcal{F}}_\theta = \left\{\phi(x) = \sum_{k=1}^{K} w_k g(x; \theta_k) \Big| w_k \geq 0, \sum_{k=1}^{K} w_k = 1\right\}. \tag{7}$$

Similar to Rahimi & Recht (2008), we consider mixtures of the form $\phi(x) = \int_\Theta w(\theta) g(x; \theta) d\theta$. For a mixture $\phi$, we define $\|\phi\|_p := \sup_\theta |\frac{w(\theta)}{p(\theta)}|$. Define

$$\mathcal{F}_p = \left\{\phi(x) = \int_\Theta w(\theta) g(x; \theta) d\theta \Big| \|\phi\|_p < \infty, w(\theta) \geq 0, \int_\Theta w(\theta) d\theta = 1\right\}. \tag{8}$$

Note that for any $\phi \in \mathcal{F}_p$, $\phi$ is a smoothed probability function. Intuitively, $\mathcal{F}_p$ is quite a rich set. The following result shows that with high probability, the best $\gamma$-robustness index a SWEEN model can obtain is near the optimal $\gamma$-robustness index in the class $\mathcal{F}_p$. Thus, the ensembling generality also holds for the $\gamma$-robustness index we defined for robustness.

**Lemma 1.** *Suppose $\gamma$ is a Lipschitz function. Given $\eta > 0$. For any $\varepsilon > 0$, for sufficently large $K$, with probability at least $1 - \eta$ over $\theta_1, ..., \theta_K$ drawn* i.i.d. *from $p$, there exists $\hat{\phi} \in \hat{\mathscr{F}}_\theta$ which satisfies*

$$\mathcal{I}_\gamma(\hat{\phi}) > \sup_{\phi \in \mathscr{F}_p} \mathcal{I}_\gamma(\phi) - \varepsilon. \tag{9}$$

*Moreover, if there exists $\phi_0 \in \mathscr{F}_p$ such that $\mathcal{I}_\gamma(\phi_0) = \sup_{\phi \in \mathscr{F}_p} \mathcal{I}_\gamma(\phi)$, $K = \Omega(\frac{1}{\varepsilon^4})$.*

In practice, the defined robustness index $\mathcal{I}_\gamma(\cdot)$ may be hard to optimized directly, in which case we choose a surrogate loss function $l : \mathbb{R}^M \times \mathcal{Y} \to \mathbb{R}$ to approximate it. Now the optimization for the ensemble weight $w$ of a SWEEN model over a training set $\{(x_i, y_i)\}_{i=1}^n$ can be formulated as

$$\min_{w \in \Delta_K} \frac{1}{n} \sum_{i=1}^n l(\sum_{k=1}^K w_k g(x_i; \theta_k), y_i). \tag{10}$$

However, this process typically invovles Monte Carlo simulation since we only have access to $f(\cdot, \theta_k), k = 1, \cdots, K$. We define the risk and empirical risk w.r.t. the surrogate loss $l$.

**Definition 2.** *(Risk and empirical risk). For a surrogate loss function $l : \mathbb{R}^M \times \mathcal{Y} \to \mathbb{R}$, the risk of a probability function $\phi$ are defined as*

$$\mathcal{R}[\phi] = \mathbb{E}_{(x,y) \sim \mathcal{D}} l(\phi(x), y). \tag{11}$$

*If $\phi(x) = \sum_{k=1}^K w_k g(x; \theta_k) \in \hat{\mathscr{F}}_\theta$, for training set $\{(x_i, y_i)\}_{i=1}^n$ and sample size $s$, the empirical risk of $\phi$ is defined as*

$$\mathcal{R}_{emp}[\phi] = \frac{1}{n} \sum_{i=1}^n l(\sum_{k=1}^K w_k [\frac{1}{s} \sum_{j=1}^s f(x_i + \delta_{ijk}; \theta_k)], y_i), \tag{12}$$

*where $\delta_{ijk} \overset{i.i.d.}{\sim} \mathcal{N}(0, \sigma^2 I), 1 \leq i \leq n, 1 \leq j \leq s, 1 \leq k \leq K$.*

Now solving for $w$ is reduced to finding the minimizer of $\mathcal{R}_{emp}$. When the loss function $l$ is convex, this problem is a low-dimensional convex optimization, so we can obtain the global empirical risk minimizer using traditional convex optimization algorithms. Furthermore, we have:

**Theorem 2.** *Suppose for all $y \in \mathcal{Y}$, $l(\cdot, y)$ is a Lipschitz function with constant $L$ and is uniformly bounded. Given $\eta > 0$. For any $\varepsilon > 0$, for sufficently large $K$, if $n = \Omega(\frac{K^2}{\varepsilon^2}), s = \Omega(\frac{\log Kn}{\varepsilon^2})$, then with probability at least $1 - \eta$ over the training dataset $\{(x_i, y_i)\}_{i=1}^n$ drawn* i.i.d. *from $\mathcal{D}$ and the parameters $\theta_1, ..., \theta_K$ drawn* i.i.d. *from $p$ and the noise samples drawn* i.i.d. *from $\mathcal{N}(0, \sigma^2 I)$, the empirical risk minimizer $\hat{\phi}$ over $\hat{\mathscr{F}}_\theta$ satisfies*

$$\mathcal{R}[\hat{\phi}] - \inf_{\phi \in \mathscr{F}_p} \mathcal{R}[\phi] < \varepsilon. \tag{13}$$

*Moreover, if there exists $\phi_0 \in \mathscr{F}_p$ such that $\mathcal{R}[\phi_0] = \inf_{\phi \in \mathscr{F}_p} \mathcal{R}[\phi]$, $K = \Omega(\frac{1}{\varepsilon^4})$.*

Theorem 2 gives a guarantee that, for large enough $K, n, s$, the gap between the risk of the empirical risk minimizer $\hat{\phi}$ and $\inf_{\phi \in \mathscr{F}_p} \mathcal{R}[\phi]$ can be arbitrarily small with high probability. Note that we can solve $\hat{\phi}$ to any given precision when $l$ is convex. Moreover, Theorem 2 reveals the accessibility of $\hat{\phi}$ in Lemma 1 when $l$ approximates the $\gamma$-robustness index well. While the number of candidate models and the number of training samples need to be large to ensure good theoretical properties, we will show that the performance of SWEEN models of practical settings is good enough in Section 5.

### 4.3 ADAPTIVE PREDICTION ALGORITHM

A major drawback of ensembling is the high execution cost during inference, which consists of prediction and certification costs for smoothed classifiers. The evaluation of smoothed classifiers relies on Monte Carlo simulation, which is computationally expensive. For instance, Cohen et al. (2019) use 100 Monte Carlo samples for prediction and 100,000 samples for certification. If we use 100 candidate models to construct a SWEEN model, the certification of a single data point will require 10,010,000 local evaluations (10,000 for prediction and 10,000,000 for certification). Inoue (2019)

observes that ensembling does not make improvements for inputs predicted with high probabilities even when they are mispredicted. He proposes an adaptive ensemble prediction algorithm to reduce the execution cost of unweighted ensemble models. We modify the algorithm to make it applicative to weighted ensemble models, which is detailed in Appendix B.1. For a data point, classifiers are evaluated in descending order with respect to their weights. Whenever an early-exit condition is satisfied, we stop the evaluation and return the current prediction.

## 5 EXPERIMENTS

In this section, we design extensive experiments on CIFAR-10, SVHN and ImageNet to evaluate the performance of SWEEN models.

### 5.1 SETUP

**Model setup**     We train different network architectures on CIFAR-10 and SVHN to serve as candidates for ensembling, including LeNet (LeCun et al., 1989), AlexNet (Krizhevsky et al., 2017), ResNet-20 (He et al., 2016), ResNet-26, ResNet-32, ResNet-110, DenseNet (Huang et al., 2017) (depth=100), VGG-16 (Simonyan & Zisserman, 2015), and VGG-19. We particularly evaluate two compositions of SWEEN models. The first is a relatively rich set of models, including LeNet, AlexNet, ResNet-20, ResNet-110, DenseNet, VGG-16, VGG-19, denoted by the SWEEN-7 model. The second is a small set of small models, including ResNet-20, ResNet-26, ResNet-32, denoted by the SWEEN-3 model. The SWEEN-7 model and the SWEEN-3 model simulate how much SWEEN can help in scenarios when we have adequate and limited numbers of candidate models, respectively. On ImageNet, we train three candidate models including ResNet-18, ResNet-34 and ResNet-50. We evaluate the SWEEN model containing the three models, denoted by the SWEEN-IN model. A SWEEN model and its candidate models are trained and evaluated with the same noise level $\sigma$. The detailed algorithm for obtaining a SWEEN model is presented in Appendix B.2.

**Candidate model Training**     We train candidate models using two training schemes, including Gaussian data augmentation training (Cohen et al., 2019), which is denoted as the standard training for simplicity, and MACER training (Zhai et al., 2020). All hyper-parameters used in our experiments are listed in Appendix C.1.

**Solving the ensembling weight**     From Section 4 we know that we can obtain the empirical risk minimizer by solving a convex optimization. However, this requires first to approximate the value of smoothed functions of candidate models at every data point, which can be very costly when the number of candidates or training data samples is large. Hence, we use Gaussian data augmented training to solve the ensembling weight. More precisely, we freeze the parameters of candidate models and minimize the cross-entropy loss of the SWEEN model on Gaussian augmented data from the evaluation set. Empirically we find that this approach is much faster and yields comparable results.

**Certification**     Following previous works, we report the *approximated certified accuracy* (ACA), which is the fraction of the test set that can be certified to be robust at radius $r$ approximately (see Cohen et al. (2019) for more details). We also report the *average certified radius* (ACR) following Zhai et al. (2020). The ACR equals to the area under the radius-accuracy curve (see Figure 1). All results were certified using algorithms in Cohen et al. (2019) with $N = 100,000$ samples and failure probability $\alpha = 0.001$.

### 5.2 RESULTS

**Standard training on CIFAR-10**     Table 1 displays the performance of two kinds of SWEEN models under noise levels $\sigma \in \{0.25, 0.50, 1.00\}$. The performance of a single ResNet-110 is included for comparison, and we also report the upper envelopes of the ACA and ACR of their corresponding candidate models as UE. In Figure 1, we display the radius-accuracy curves for the SWEEN models and all their corresponding candidate models under $\sigma = 0.50$ on CIFAR-10. We also include full-size figures in Appendix D.

The results show that SWEEN models significantly boost the performance compared to their corresponding candidate models. According to Figure 1, the SWEEN-7 model consistently outperforms

Table 1: ACA (%) and ACR on CIFAR-10. All models are trained via the standard training. UE stands for the upper envelope, which shows the largest ACA and ACR among the candidate models.

| $\sigma$ | Model | 0.00 | 0.25 | 0.5 | 0.75 | 1.00 | 1.25 | 1.50 | 1.75 | 2.00 | ACR |
|---|---|---|---|---|---|---|---|---|---|---|---|
| | ResNet-110 | 79.6 | 65.2 | 50.8 | 34.4 | 0 | 0 | 0 | 0 | 0 | 0.489 |
| | UE$_{\text{SWEEN-3}}$ | 80.5 | 65.6 | 47.9 | 30.2 | 0 | 0 | 0 | 0 | 0 | 0.470 |
| 0.25 | SWEEN-3 | 82.3 | 69.8 | 54.7 | 35.9 | 0 | 0 | 0 | 0 | 0 | 0.520 |
| | UE$_{\text{SWEEN-7}}$ | 80.5 | 67.9 | 52.2 | 36.1 | 0 | 0 | 0 | 0 | 0 | 0.506 |
| | SWEEN-7 | 84.2 | 72.0 | 58.7 | 43.0 | 0 | 0 | 0 | 0 | 0 | **0.560** |
| | ResNet-110 | 68.7 | 58.6 | 46.7 | 35.4 | 25.0 | 17.0 | 9.0 | 4.6 | 0 | 0.573 |
| | UE$_{\text{SWEEN-3}}$ | 69.6 | 58.3 | 45.8 | 33.7 | 23.0 | 15.8 | 9.2 | 4.7 | 0 | 0.556 |
| 0.50 | SWEEN-3 | 70.9 | 61.4 | 50.8 | 38.3 | 27.7 | 20.1 | 12.8 | 6.7 | 0 | 0.630 |
| | UE$_{\text{SWEEN-7}}$ | 68.6 | 58.9 | 46.6 | 34.8 | 24.7 | 16.5 | 10.2 | 5.3 | 0 | 0.574 |
| | SWEEN-7 | 71.2 | 63.0 | 52.2 | 41.9 | 31.2 | 22.9 | 15.3 | 8.3 | 0 | **0.678** |
| | ResNet-110 | 51.4 | 44.9 | 37.9 | 31.8 | 24.6 | 18.8 | 13.8 | 10.2 | 6.7 | 0.559 |
| | UE$_{\text{SWEEN-3}}$ | 50.6 | 44.7 | 38.2 | 30.8 | 24.6 | 18.5 | 13.6 | 10.5 | 7.0 | 0.555 |
| 1.00 | SWEEN-3 | 51.9 | 45.5 | 39.3 | 32.3 | 25.9 | 19.7 | 15.4 | 11.4 | 8.1 | 0.595 |
| | UE$_{\text{SWEEN-7}}$ | 52.0 | 45.7 | 37.9 | 31.9 | 25.1 | 19.2 | 13.9 | 10.1 | 7.2 | 0.557 |
| | SWEEN-7 | 52.7 | 46.3 | 39.8 | 34.0 | 27.6 | 22.7 | 17.9 | 12.6 | 9.2 | **0.631** |

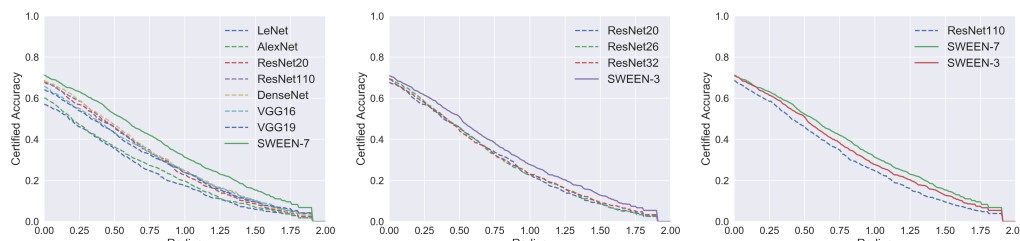

Figure 1: Radius-accuracy curves under $\sigma = 0.50$. All models are trained via the standard training. (**Left**) The SWEEN-7 model and all its candidate models. (**Middle**) The SWEEN-3 model and all its candidate models. (**Right**) The SWEEN-7 model, the SWEEN-3 model and the ResNet-110.

all its candidates in terms of the ACA at all radii. The ACR of the SWEEN-7 model is 0.678, much higher than that of the upper envelope of the candidates, which is 0.574. It confirms our theoretical analysis in Section 4 that SWEEN can combine the strength of candidate models and attain superior performance. Besides, SWEEN is effective when only limited numbers of small candidate models are available. The SWEEN-3 model using ResNet-20, ResNet-26, and ResNet-32 achieves higher ACA than the ResNet-110 at all radii on all noise levels. The total training time and the number of parameters of the SWEEN-3 model are 36 % and 30% less than those of ResNet-110, respectively. The improvements can be further amplified by increasing the number and size of candidate models. As an instance, the ACR of the SWEEN-7 model is at least 13% higher than that of the ResNet-110 in Table 1. The above results verify the effectiveness of SWEEN for randomized smoothing.

Table 2: Training time, #parameters and #FLOPs for models under $\sigma = 0.50$ via MACER training. All the experiments are run on a single NVIDIA 1080 Ti GPU.

| Model | sec/epoch | #epochs | Total hrs | #parameters | #FLOPs |
|---|---|---|---|---|---|
| ResNet-110 | 404.2 | 440 | 49.4 | 1.73M | 255.27M |
| ResNet-20 | 72.2 | 440 | 8.8 | 0.27M | 41.21M |
| ResNet-26 | 92.9 | 440 | 11.3 | 0.36M | 55.48M |
| ResNet-32 | 113.2 | 440 | 13.8 | 0.46M | 69.75M |
| Weight | 0.6 | 150 | 0.025 | - | - |
| Ensemble | - | - | 33.9 | 1.10M | 166.44M |

Table 3: ACA (%) and ACR on CIFAR-10. All models are trained via MACER training. UE stands for the upper envelope of candidate models.

| $\sigma$ | Model | 0.00 | 0.25 | 0.5 | 0.75 | 1.00 | 1.25 | 1.50 | 1.75 | 2 | ACR |
|---|---|---|---|---|---|---|---|---|---|---|---|
| 0.25 | ResNet-110 | 81 | 71 | 59 | 43 | 0 | 0 | 0 | 0 | 0 | 0.556 |
| | UE$_{SWEEN-3}$ | 77.4 | 66.9 | 56.8 | 41.9 | 0 | 0 | 0 | 0 | 0 | 0.529 |
| | SWEEN-3 | 77.7 | 68.7 | 60.3 | 46.6 | 0 | 0 | 0 | 0 | 0 | **0.558** |
| 0.50 | ResNet-110 | 66 | 60 | 53 | 46 | 38 | 29 | 19 | 12 | 0 | 0.726 |
| | UE$_{SWEEN-3}$ | 64.9 | 57.1 | 49.7 | 41.1 | 34.1 | 26.2 | 20.2 | 11.7 | 0 | 0.685 |
| | SWEEN-3 | 64.7 | 58.4 | 51.8 | 43.9 | 37.2 | 29.2 | 22.8 | 14.6 | 0 | **0.727** |
| 1.00 | ResNet-110 | 45 | 41 | 38 | 35 | 32 | 29 | 25 | 22 | 18 | 0.792 |
| | UE$_{SWEEN-3}$ | 39.4 | 38.2 | 35.8 | 33.4 | 30.3 | 27.6 | 24.5 | 22.0 | 19.0 | 0.793 |
| | SWEEN-3 | 39.5 | 37.9 | 35.8 | 33.2 | 30.4 | 27.5 | 24.6 | 22.0 | 19.1 | **0.796** |

**MACER training on CIFAR-10**    Since SWEEN is compatible with previous training algorithms, we adopt MACER training for the SWEEN-3 model. The results are summarized in Table 3. For the ACA and ACR of the ResNet-110 model, we use the original numbers from Zhai et al. (2020).

In the results, the SWEEN-3 model achieves comparable results to the ResNet-110 but is more efficient. From Table 2 and 3, the SWEEN-3 model takes 33.9 hours to achieve 0.727 in terms of the ACR for $\sigma = 0.5$, using three small and easy-to-train candidates models. Meanwhile, it takes 49.4 hours for the ResNet-110 to achieve similar performance on CIFAR-10. This 32% speed up reveals the efficiency of applying SWEEN to previous training methods.

**Standard training on ImageNet**    Table 4 displays the performance of ResNet-50 and SWEEN-IN under the noise levels $\sigma \in \{0.25, 0.50, 1.00\}$. We note that the performance of ResNet-50 is also the upper envelope of the three models in SWEEN-IN. We can see that the SWEEN-IN model significantly outperforms its corresponding candidate models, similar to the results on CIFAR-10. The results again confirm the effectiveness of our proposed SWEEN framework.

Table 4: ACA (%) and ACR on ImageNet. All models are trained via standard training.

| $\sigma$ | Model | 0.00 | 0.25 | 0.5 | 0.75 | 1.00 | 1.25 | 1.50 | 1.75 | 2 | ACR |
|---|---|---|---|---|---|---|---|---|---|---|---|
| 0.25 | ResNet-50 | 66.8 | 58.2 | 49.0 | 38.2 | 0 | 0 | 0 | 0 | 0 | 0.469 |
| | SWEEN-IN | **67.8** | **60.2** | **51.6** | **41.6** | 0 | 0 | 0 | 0 | 0 | **0.489** |
| 0.50 | ResNet-50 | **56.4** | **52.4** | 46.4 | 42.2 | 37.8 | 32.6 | 28.0 | 21.4 | 0 | 0.726 |
| | SWEEN-IN | **56.4** | **52.4** | **49.6** | **45.0** | **40.8** | **37.0** | **31.6** | **25.2** | 0 | **0.781** |
| 1.00 | ResNet-50 | 43.6 | 40.6 | 37.8 | 35.4 | 32.4 | 28.8 | 25.8 | 22.4 | 19.4 | 0.863 |
| | SWEEN-IN | **44.6** | **42.0** | **38.6** | **36.4** | **35.0** | **32.6** | **29.4** | **25.6** | **22.4** | **0.948** |

**Other experimental results**    Due to space constraints, we only report the main results here. The results of further experiments (e.g., the results on SVHN, the results of the adaptive prediction algorithm, the results of the SWEEN model with identical architectures, and the results of SWEEN versus real attacks) can be found in Appendix C.

# 6    CONCLUSIONS

In this work, we introduced the smoothed weighted ensembling (SWEEN) to improve randomized smoothed classifiers in terms of both accuracy and robustness. We showed that SWEEN can achieve optimal certified robustness w.r.t. our defined $\gamma$-robustness index. Furthermore, we can obtain the optimal SWEEN model w.r.t. a surrogate loss from training under mild assumptions. We also developed an adaptive prediction algorithm to accelerate the prediction and certification process. Our extensive experiments showed that a properly designed SWEEN model was able to outperform all its candidate models by a significant margin consistently. Moreover, SWEEN models using a few small and easy-to-train candidates could match or exceed a large individual model on performance with a notable reduction in total training time. Our theoretical and empirical results confirmed that SWEEN is a viable tool for improving the performance of randomized smoothing models.

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

## A  PROOFS

### A.1  PROOF OF LEMMA 1

Define

$$\mathscr{F}'_p = \left\{ \phi(x) = \int_\Theta w(\theta) g(x; \theta) d\theta \Big| \; \|\phi\|_p < \infty, w(\theta) \geq 0 \right\}, \tag{14}$$

$$\hat{\mathscr{F}}'_\theta = \left\{ \phi(x) = \sum_{k=1}^K w_k g(x; \theta_k) \Big| w_k \geq 0 \right\}. \tag{15}$$

We have $\mathscr{F}_p \subseteq \mathscr{F}'_p, \hat{\mathscr{F}}_\theta \subseteq \hat{\mathscr{F}}'_\theta$.

**Lemma 2.** *Let $\mu$ be any probability measure on $\mathbb{R}^d$. For $\phi : \mathbb{R}^d \to \mathbb{R}^M$, define the norm $\|\phi\|_\mu^2 \triangleq \int_{\mathbb{R}^d} \|\phi(x)\|_2^2 d\mu(x)$. Fix $\phi \in \mathscr{F}'_p$, then for any $\eta > 0$, with probability at least $1 - \eta$ over $\theta_1, ..., \theta_K$ drawn i.i.d. from $p$, there exists $\hat{\phi}(x) = \sum_{k=1}^K c_k g(x; \theta_k) \in \hat{\mathscr{F}}'_\theta$ which satisfies*

$$\|\hat{\phi} - \phi\|_\mu \leq \frac{\|\phi\|_p}{\sqrt{K}} \left( 1 + \sqrt{2 \log \frac{1}{\eta}} \right). \tag{16}$$

*Proof.* Sine $\phi \in \mathscr{F}'_p$, we can write $\phi(x) = \int_\Theta w(\theta) g(x; \theta) d\theta$, where $w(\theta) \geq 0$. Construct $\phi_k = \beta_k g(\cdot; \theta_k), k = 1, 2, \cdots, K$, where $\beta_k = \frac{w(\theta_k)}{p(\theta_k)}$, then $\mathbb{E}\phi_k = \phi, \|\phi_k\|_\mu = \sqrt{\int_{\mathbb{R}^d} \beta_k^2 \|g(x; \theta_k)\|_2^2 d\mu(x)} \leq |\beta_k| \leq \|\phi\|_p$. We then define

$$u(\theta_1, \cdots, \theta_K) = \|\frac{1}{K} \sum_{k=1}^K \phi_k - \phi\|_\mu. \tag{17}$$

First, by using Jensen's inequality and the fact that $\|\phi_k\|_\mu \le \|\phi\|_p$, we have

$$\mathbb{E}[u(\theta)] \le \sqrt{\mathbb{E}[u^2(\theta)]} = \sqrt{\mathbb{E}[\|\frac{1}{K}\sum_{k=1}^{K}\phi_k - \mathbb{E}\phi_k\|_\mu^2]} = \sqrt{\frac{1}{K}(\mathbb{E}\|\phi_k\|_\mu^2 - \|\mathbb{E}\phi_k\|_\mu^2)} \le \frac{\|\phi\|_p}{\sqrt{K}}.$$

Next, for $\theta_1, \cdots, \theta_M$ and $\tilde\theta_i$, we have

$$|u(\theta_1, \cdots, \theta_M) - u(\theta_1, \cdots, \tilde\theta_i, \cdots, \theta_M)|$$

$$= \ |\|\frac{1}{K}\sum_{k=1}^{K}\phi_k - \phi\|_\mu - \|\frac{1}{K}(\sum_{k=1,k\neq i}^{K}\phi_k + \tilde\phi_i) - \phi\|_\mu|$$

$$\le \ \|\frac{1}{K}\sum_{k=1}^{M}\phi_k - \frac{1}{K}(\sum_{k=1,k\neq i}^{M}\phi_k + \tilde\phi_i)\|_\mu$$

$$= \ \frac{\|\phi_i - \tilde\phi_i\|_\mu}{K}$$

$$\le \ \frac{2\|\phi\|_p}{K}.$$

Now we can use McDiarmid's inequality to bound $u(\theta)$, which gives

$$\mathbb{P}[u(\theta) - \frac{\|\phi\|_p}{\sqrt{K}} \ge \varepsilon] \le \mathbb{P}[u(\theta) - \mathbb{E}u(\theta) \ge \varepsilon] \le \exp(-\frac{K\varepsilon^2}{2\|\phi\|_p^2}). \tag{18}$$

The theorem follows by setting $\delta$ to the right hand side and solving $\varepsilon$.

$\square$

**Lemma 3.** *Let $\mu$ be any probability measure on $\mathbb{R}^d$. For $\phi : \mathbb{R}^d \to \mathbb{R}^M$, define the norm $\|\phi\|_\mu^2 \triangleq \int_{\mathbb{R}^d} \|\phi(x)\|_2^2 d\mu(x)$, then for any $\eta > 0$, for $K \ge M\|\phi\|_p^2(1 + \sqrt{2\log\frac{1}{\eta}})^2$, with probability at least $1 - \eta$ over $\theta_1, ..., \theta_K$ drawn* i.i.d. *from $p$, there exists $\hat\phi(x) = \sum_{k=1}^{K} c_k g(x; \theta_k) \in \hat{\mathscr{F}}_\theta$ which satisfies*

$$\|\hat\phi - \phi\|_\mu < 2\sqrt{\|\phi\|_p}\sqrt[4]{\frac{M}{K}}(1 + \sqrt{2\log\frac{1}{\eta}})^{\frac{1}{2}}. \tag{19}$$

*Proof.* Fix $\phi \in \mathscr{F}_p \subseteq \mathscr{F}'_p$, by using Lemma 2, we have that for any $\delta > 0$, with probability at least $1 - \eta$ over $\theta_1, ..., \theta_K$ drawn *i.i.d.* from $p$, there exists $\tilde\phi(x) = \sum_{k=1}^{K} c_k g(x; \theta_k) \in \hat{\mathscr{F}}'_\theta$ which satisfies

$$\|\tilde\phi - \phi\|_\mu < \frac{\|\phi\|_p}{\sqrt{K}}(1 + \sqrt{2\log\frac{1}{\eta}}) \triangleq B(K). \tag{20}$$

Denote $C = \sum_{k=1}^{K} c_k$, and define $s(t) \triangleq \sum_{i=1}^{M} t_i$ as the sum of all elements of $t \in \mathbb{R}^M$. Then $s(g(x; \theta)) = 1, \forall x \in \mathbb{R}^d, \theta \in \Theta$. Thus,

$$s(\phi(x)) = \sum_{i=1}^{M} \phi_i(x) = \sum_{i=1}^{M} \int_\Theta w(\theta) g_i(x; \theta) d\theta = \int_\Theta w(\theta) \sum_{i=1}^{M} g_i(x; \theta) d\theta = \int_\Theta w(\theta) d\theta = 1,$$

$$s(\tilde\phi(x)) = \sum_{i=1}^{M} \tilde\phi_i(x) = \sum_{i=1}^{M}\sum_{k=1}^{K} c_k g_i(x; \theta_k) = \sum_{k=1}^{K} c_k \sum_{i=1}^{M} g_i(x; \theta_k) = \sum_{k=1}^{K} c_k = C.$$

Now we have

$$
\begin{aligned}
B(K)^2 > \|\tilde{\phi} - \phi\|_\mu^2 &= \int_{\mathbb{R}^d} \|\tilde{\phi}(x) - \phi(x)\|_2^2 d\mu(x) \\
&\geq \int_{\mathbb{R}^d} \frac{(s(\tilde{\phi}(x) - \phi(x)))^2}{M} d\mu(x) \\
&= \int_{\mathbb{R}^d} \frac{(C-1)^2}{M} d\mu(x) \\
&= \frac{(C-1)^2}{M},
\end{aligned}
$$

which gives $1 - \sqrt{M}B(K) < C < 1 + \sqrt{M}B(K)$. Construct $\hat{\phi}(x) = \frac{\tilde{\phi}(x)}{C}$, then $\hat{\phi} \in \hat{\mathscr{F}}_\theta$ and

$$
\begin{aligned}
\|\hat{\phi} - \phi\|_\mu^2 &= \int_{\mathbb{R}^d} \|\hat{\phi}(x) - \phi(x)\|_2^2 d\mu(x) \\
&= \int_{\mathbb{R}^d} \|C^{-1}\tilde{\phi}(x) - \phi(x)\|_2^2 d\mu(x) \\
&= \int_{\mathbb{R}^d} \|(\tilde{\phi}(x) - \phi(x)) + (C^{-1} - 1)\tilde{\phi}(x)\|_2^2 d\mu(x) \\
&= \int_{\mathbb{R}^d} (\|\tilde{\phi}(x) - \phi(x)\|_2^2 + \|(C^{-1} - 1)\tilde{\phi}(x)\|_2^2 + 2(C^{-1} - 1)\langle\tilde{\phi}(x) - \phi(x), \tilde{\phi}(x)\rangle) d\mu(x) \\
&= \int_{\mathbb{R}^d} (\|\hat{\phi}(x) - \phi(x)\|_2^2 + (C^{-2} - 1)\|\tilde{\phi}(x)\|_2^2 + 2(1 - C^{-1})\langle\phi(x), \tilde{\phi}(x)\rangle) d\mu(x).
\end{aligned}
$$

Sine we have $\frac{C^2}{M} \leq \|\tilde{\phi}(x)\|_2^2 \leq C^2$, $|\langle\phi(x), \tilde{\phi}(x)\rangle| \leq \sqrt{\|\phi(x)\|_2^2\|\tilde{\phi}(x)\|_2^2} \leq C$, it holds that

(i) When $1 < C < 1 + \sqrt{M}B(K)$,

$$
\begin{aligned}
\|\hat{\phi} - \phi\|_\mu^2 &\leq \int_{\mathbb{R}^d} (\|\tilde{\phi}(x) - \phi(x)\|_2^2 + \frac{1 - C^2}{M} + 2(C - 1)) d\mu(x) \\
&\leq B(K)^2 + \frac{1 - C^2}{M} + 2(C - 1) \\
&\leq -\frac{2B(K)}{\sqrt{M}} + 2\sqrt{M}B(K);
\end{aligned}
$$

(ii) When $1 - \sqrt{M}B(K) < C \leq 1$,

$$
\begin{aligned}
\|\hat{\phi} - \phi\|_\mu^2 &\leq \int_{\mathbb{R}^d} (\|\tilde{\phi}(x) - \phi(x)\|_2^2 + (1 - C^2) + 2(1 - C)) d\mu(x) \\
&\leq B(K)^2 + 4 - (1 + C)^2 \\
&\leq 4\sqrt{M}B(K) - (M - 1)B(K)^2 \\
&< 4\sqrt{M}B(K).
\end{aligned}
$$

Thus, with probability at least $1 - \eta$ over $\theta_1, ..., \theta_K$ drawn *i.i.d.* from $p$,

$$
\|\hat{\phi} - \phi\|_\mu < 2\sqrt{\sqrt{M}B(K)} = 2\sqrt{\|\phi\|_p} \sqrt[4]{\frac{M}{K}}(1 + \sqrt{2\log\frac{1}{\eta}})^{\frac{1}{2}}.
$$

$\square$

**Lemma 4.** *Suppose $l(\cdot, \cdot)$ is $L$-Lipschitz in its first argument. Fix $\phi \in \mathscr{F}_p$, then for any $\eta > 0$, for $K \geq M\|\phi\|_p^2(1 + \sqrt{2\log\frac{1}{\eta}})^2$, with probability at least $1 - \eta$ over $\theta_1, ..., \theta_K$ drawn i.i.d. from $p$, there exists $\hat{\phi} \in \hat{\mathscr{F}}_\theta$ which satisfies*

$$
|\mathbb{E}_{(x,y)\sim\mathcal{D}}[l(\hat{\phi}(x), y)] - \mathbb{E}_{(x,y)\sim\mathcal{D}}[l(\phi(x), y)]| < 2L\sqrt{\|\phi\|_p} \sqrt[4]{\frac{M}{K}}(1 + \sqrt{2\log\frac{1}{\eta}})^{\frac{1}{2}}.
$$

*Proof.*

$$
\begin{aligned}
|\mathbb{E}[l(\hat{\phi}(x), y)] - \mathbb{E}[l(\phi(x), y)]| &\leq \mathbb{E}|c(\phi(x), y) - c(\hat{\phi}(x), y)| \\
&\leq L\mathbb{E}\|\phi(x) - \hat{\phi}(x)\|_2 \\
&\leq L\sqrt{\mathbb{E}\|\phi(x) - \hat{\phi}(x)\|_2^2} \\
&= L\|\phi - \hat{\phi}\|_{\mathcal{D}|_x}
\end{aligned}
$$

The desired result follows from Lemma 3. $\qquad\square$

**Lemma 5.** *(Corollary of Proposition 1 in Zhai et al. (2020)) Given any $p_1, p_2, \cdots, p_M$ satisfies $p_1 \geq p_2 \geq \cdots \geq p_M \geq 0$ and $p_1 + p_2 + \cdots + p_M = 1$. The derivative of $\text{clip}(\frac{\sigma}{2}[\Phi^{-1}(p_1) - \Phi^{-1}(p_2)]; 0, D)$ with respect to $p_1$ and $p_2$ is bounded.*

Now we can prove Lemma 1.

*Proof of Lemma 1.* Let $\phi_0 \in \mathscr{F}_p$ such that $\mathcal{I}_\gamma(\phi_0) > \sup_{\phi \in \mathscr{F}_p} \mathcal{I}_\gamma(\phi) - \frac{\varepsilon}{2}$. From Lemma 5 we know that $q(p, y) \triangleq \text{clip}(\frac{\sigma}{2}[\Phi^{-1}(p_y) - \Phi^{-1}(\max_{k \neq y} p_k)]; 0, D)$ is Lipschitz in its first argument. Since $m$ is Lipschitz, $c(p, y) \triangleq m(q(p, y))$ is also Lipschitz in its first argument with some constant $L$. Apply Lemma 4, we have that for $K \geq M\|\phi\|_p^2(1 + \sqrt{1 + 2\log\frac{1}{\delta}})^2$, with probability at least $1 - \eta$ over $\theta_1, ..., \theta_K$ drawn *i.i.d.* from $p$, there exists $\hat{\phi} \in \hat{\mathscr{F}}_\theta$ which satisfies

$$
\begin{aligned}
\mathcal{I}_\gamma(\phi_0) - \mathcal{I}_\gamma(\hat{\phi}) &= \mathbb{E}_{(x,y)\sim\mathcal{D}}[l(\phi_0(x), y)] - \mathbb{E}_{(x,y)\sim\mathcal{D}}[l(\hat{\phi}(x), y)] \\
&< 2L\sqrt{\|\phi_0\|_p}\sqrt[4]{\frac{M}{K}}(1 + \sqrt{2\log\frac{1}{\eta}})^{\frac{1}{2}}.
\end{aligned}
$$

When $K > \frac{256L^4\|\phi_0\|_p^2 M(1 + \sqrt{2\log\frac{1}{\eta}})^2}{\varepsilon^4}$, we have

$$
\sup_{\phi \in \mathscr{F}_p} \mathcal{I}_\gamma(\phi) - \mathcal{I}_\gamma(\hat{\phi}) = (\sup_{\phi \in \mathscr{F}_p} \mathcal{I}_\gamma(\phi) - \mathcal{I}_\gamma(\phi_0)) + (\mathcal{I}_\gamma(\phi_0) - \mathcal{I}_\gamma(\hat{\phi})) < \frac{\varepsilon}{2} + \frac{\varepsilon}{2} = \varepsilon.
$$

If $\mathcal{I}_\gamma(\phi_0) = \sup_{\phi \in \mathscr{F}_p} \mathcal{I}_\gamma(\phi)$, which means $\|\phi_0\|_p$ is independent of $\varepsilon$, $K = \Omega(\frac{1}{\varepsilon^4})$. $\qquad\square$

## A.2  PROOF OF THEOREM 2

First we introduce some results from statistical learning theory.

**Definition 3.** *(Gaussian complexity). Let $\mu$ be a probability distribution on a set $\mathcal{X}$ and suppose that $x_1, ..., x_n$ are independent samples selected according to $\mu$. Let $\mathscr{F}$ be a class of functions mapping from $X$ to $\mathbb{R}$. The Gaussian complexity of $\mathscr{F}$ is*

$$
G_n[\mathscr{F}] \triangleq \mathbb{E}[\sup_{f \in \mathscr{F}} |\frac{2}{n}\sum_{i=1}^n \xi_i f(x_i)| \big| x_1, ..., x_n; \xi_1, ..., \xi_n]
$$

*where $\xi_1, ..., \xi_n$ are independent $\mathcal{N}(0, 1)$ random variables.*

**Definition 4.** *(Rademacher complexity) Let $\mu$ be a probability distribution on a set $\mathcal{X}$ and suppose that $x_1, ..., x_n$ are independent samples selected according to $\mu$. Let $\mathscr{F}$ be a class of functions mapping from $X$ to $\mathbb{R}$. The Rademacher complexity of $\mathscr{F}$ is*

$$
R_n[\mathscr{F}] \triangleq \mathbb{E}[\sup_{f \in \mathscr{F}} |\frac{2}{n}\sum_{i=1}^n \sigma_i f(x_i)| \big| x_1, ..., x_n; \sigma_1, ..., \sigma_n]
$$

*where $\sigma_1, ..., \sigma_n$ are independent uniform $\{\pm 1\}$-valued random variables.*

**Lemma 6.** *(Part of Lemma 4 in Bartlett & Mendelson (2001)). There are absolute constants $\beta$ such that for every class $\mathscr{F}$ and every integer $n$, $R_n(\mathscr{F}) \leq \beta G_n(\mathscr{F})$.*

**Lemma 7.** *(Corollary of Theorem 8 in Bartlett & Mendelson (2001)). Consider a loss function $c : \mathcal{A} \times Y \to [0, 1]$. Let $\mathscr{F}$ be a class of functions mapping from $\mathcal{X}$ to $\mathcal{A}$ and let $(x_i, y_i)_{i=1}^{n}$ be independently selected according to the probability measure $\mu$. Then, for any integer $n$ and any $0 < \eta < 1$, with probability at least $1 - \eta$ over samples of length $n$, every $f$ in $\mathscr{F}$ satisfies*

$$\mathbb{E}_{(x,y)\sim\mu}[c(f(x), y)] \leq \frac{1}{n} \sum_{i=1}^{n} c(f(x_i), y_i) + R_n[\tilde{c} \circ \mathscr{F}] + \sqrt{\frac{8 \log \frac{2}{\eta}}{n}},$$

*where $\tilde{c} \circ \mathscr{F} = \{(x, y) \mapsto c(f(x), y) - c(0, y) \big| f \in \mathscr{F}\}$*

**Lemma 8.** *(Corollary of Theorem 14 in Bartlett & Mendelson (2001)). Let $\mathcal{A} = \mathbb{R}^M$ and let $\mathscr{F}$ be a class of functions mapping from $\mathcal{X}$ to $\mathcal{A}$. Suppose that there are real-valued classes $\mathscr{F}_1, ..., \mathscr{F}_M$ such that $\mathscr{F}$ is a subset of their Cartesian product. Assume further that $c : \mathcal{A} \times Y \to \mathbb{R}$ is such that, for all $y \in Y$, $c(\cdot, y)$ is a Lipschitz function with constant $L$ which passes through the origin and is uniformly bounded. Then*

$$G_n(c \circ \mathscr{F}) \leq 2L \sum_{i=1}^{M} G_n(\mathscr{F}_i).$$

Now we prove the following lemma:

**Lemma 9.** *Let $c, \mathscr{F}, (x_i, y_i)_{i=1}^{n}, \tilde{c} \circ \mathscr{F}$ be as in Lemma 7. Then, for any integer $n$ and any $0 < \eta < 1$, with probability at least $1 - \eta$ over samples of length $n$, every $f$ in $\mathscr{F}$ satisfies*

$$\frac{1}{n} \sum_{i=1}^{n} c(f(x_i), y_i) \leq \mathbb{E}_{(x,y)\sim\mu}[c(f(x), y)] + R_n[\tilde{c} \circ \mathscr{F}] + \sqrt{\frac{8 \log \frac{2}{\eta}}{n}}.$$

*Proof.*

$$
\frac{1}{n} \sum_{i=1}^{n} c(f(x_i), y_i) - \mathbb{E}_{(x,y)\sim\mu}[c(f(x), y)] \quad \leq \quad \sup_{h \in co\mathscr{F}} (\hat{\mathbb{E}}_n h - \mathbb{E}h)
$$

$$
= \quad \sup_{h \in \tilde{c}o\mathscr{F}} (\hat{\mathbb{E}}_n h - \mathbb{E}h) + \hat{\mathbb{E}}_n c(0, y) - \mathbb{E}c(0, y).
$$

When an $(x_i, y_i)$ pair changes, the random variable $\sup_{h \in \tilde{c}o\mathscr{F}} (\hat{\mathbb{E}}_n h - \mathbb{E}h)$ can change by no more than $\frac{2}{n}$. McDiarmid's inequality implies that with probability at least $1 - \frac{\eta}{2}$,

$$\sup_{h \in \tilde{c}o\mathscr{F}} (\hat{\mathbb{E}}_n h - \mathbb{E}h) \leq \mathbb{E} \sup_{h \in \tilde{c}o\mathscr{F}} (\hat{\mathbb{E}}_n h - \mathbb{E}h) + \sqrt{\frac{2 \log \frac{2}{\eta}}{n}}.$$

A similar argument, together with the fact that $\mathbb{E}\hat{\mathbb{E}}_n c(0, y) = \mathbb{E}c(0, y)$, shows that with probability at least $1 - \eta$,

$$\mathbb{R}_{emp}[f] \leq \mathbb{R}[f] + \mathbb{E} \sup_{h \in \tilde{c}o\mathscr{F}} (\hat{\mathbb{E}}_n h - \mathbb{E}h) + \sqrt{\frac{8 \log \frac{2}{\eta}}{n}}.$$

It's left to show that $\mathbb{E} \sup_{h \in \tilde{c}o\mathscr{F}} (\hat{\mathbb{E}}_n h - \mathbb{E}h) \leq \mathcal{R}_n[\tilde{c} \circ \mathscr{F}]$. Let $(x_1', y_1'), ..., (x_n', y_n')$ be drawn *i.i.d.* from $\mu$ and independent from $(x_i, y_i)_{i=1}^{n}$, then

$$
\mathbb{E} \sup_{h \in \tilde{c}o\mathscr{F}} (\hat{\mathbb{E}}_n h - \mathbb{E}h) \quad = \quad \mathbb{E} \sup_{h \in \tilde{c}o\mathscr{F}} \mathbb{E}[\hat{\mathbb{E}}_n h - \frac{1}{n} \sum_{i=1}^{n} h(x_i', y_i')]
$$

$$
\leq \quad \mathbb{E}\mathbb{E} \sup_{h \in \tilde{c}o\mathscr{F}} [\hat{\mathbb{E}}_n h - \frac{1}{n} \sum_{i=1}^{n} h(x_i', y_i')]
$$

$$
= \quad \mathbb{E} \sup_{h \in \tilde{c}o\mathscr{F}} \frac{1}{n}(\sum_{i=1}^{n} h(x_i, y_i) - \sum_{i=1}^{n} h(x_i', y_i'))
$$

$$
\leq \quad 2\mathbb{E} \sup_{h \in \tilde{c}o\mathscr{F}} \frac{1}{n} \sum_{i=1}^{n} \sigma_i h(x_i, y_i)
$$

$$
\leq \quad \mathcal{R}_n[\tilde{c} \circ \mathscr{F}].
$$

□

We can prove the following result:

**Theorem 3.** *Let $\mathcal{A} = \mathbb{R}^M$ and let $\mathscr{F}$ be a class of functions mapping from $\mathcal{X}$ to $\mathcal{A}$. Suppose that there are real-valued classes $\mathscr{F}_1, ..., \mathscr{F}_M$ such that $\mathscr{F}$ is a subset of their Cartesian product. Assume further that the loss function $c : \mathcal{A} \times Y \to \mathbb{R}$ is such that, for all $y \in Y$, $c(\cdot, y)$ is a Lipschitz function with constant $L$ and is uniformly bounded. Let $\{(x_i, y_i)\}_{i=1}^n$ be independently selected according to the probability measure $\mu$. Then, for any integer $n$ and any $0 < \eta < 1$, there is a probability of at least $1 - \eta$ that every $f \in \mathscr{F}$ has*

$$|\frac{1}{n}\sum_{i=1}^n c(f(x_i), y_i) - \mathbb{E}_{(x,y)\sim\mu}[c(f(x), y)]| \leq \beta L \sum_{j=1}^M G_n[\mathscr{F}_j] + \sqrt{\frac{8\log\frac{4}{\eta}}{n}},$$

*where $\beta$ is a constant.*

*Proof.* From Lemma 7 and 9 we have that with probability at least $1 - \eta$ over samples of length $n$, every $f$ in $\mathscr{F}$ satisfies

$$|\frac{1}{n}\sum_{i=1}^n c(f(x_i), y_i) - \mathbb{E}_{(x,y)\sim\mu}[c(f(x), y)]| \leq R_n[\tilde{c} \circ \mathscr{F}] + \sqrt{\frac{8\log\frac{4}{\eta}}{n}},$$

it follows by applying Lemma 6 and 8. □

**Lemma 10.** *Let $c(\cdot, \cdot), \beta$ be as in Theorem 3. Let $(x_i, y_i)_{i=1}^n$ be independently selected according to the probability measure $\mathcal{D}$. For any integer $n$ and any $0 < \eta < 1$, there is a probability of at least $1 - \eta$ that every $f \in \hat{\mathscr{F}}_\theta$ has*

$$|\frac{1}{n}\sum_{i=1}^n c(f(x_i), y_i) - \mathbb{E}_{(x,y)\sim\mathcal{D}}[c(f(x), y)]| \leq \frac{2\beta LMK}{\sqrt{n}} + \sqrt{\frac{8\log\frac{4}{\eta}}{n}}.$$

*Proof.* Denote

$$\hat{\mathscr{F}}_\theta(i) = \left\{\phi(x) = \sum_{k=1}^K w_k g_i(x; w_k) \Big| w_k \geq 0, \sum_{k=1}^K w_k = 1\right\}, 1 \leq i \leq M.$$

We have that $\hat{\mathscr{F}}_\theta \subseteq \bigotimes_{k=1}^{M} \hat{\mathscr{F}}_\theta(i)$, where $\bigotimes$ stands for a Cartesian product operation. The Gaussian comlexities of $\hat{\mathscr{F}}_\theta(i)$'s can be bounded as

$$
\begin{aligned}
G_n[\hat{\mathscr{F}}_\theta(j)] &= \mathbb{E}_{x,\xi}[\sup_{\phi \in \hat{\mathscr{F}}_\theta(j)} |\frac{2}{n}\sum_{i=1}^{n}\xi_i\phi(x_i)|] \\
&= \mathbb{E}_{x,\xi}[\sup_{w}|\frac{2}{n}\sum_{i=1}^{n}\xi_i\sum_{k=1}^{K}w_kg_j(x_i;w_k)|] \\
&= \mathbb{E}_{x,\xi}[\sup_{w}|\sum_{k=1}^{K}w_k\frac{2}{n}\sum_{i=1}^{n}\xi_ig_j(x_i;w_k)|] \\
&\leq \mathbb{E}_{x,\xi}[2\sum_{k=1}^{K}|\frac{1}{n}\sum_{i=1}^{n}\xi_ig_j(x_i;w_k)|] \\
&\leq \mathbb{E}_{x}[2\sum_{k=1}^{K}\sqrt{\mathbb{E}_g(\frac{1}{n}\sum_{i=1}^{n}\xi_ig_j(x_i;w_k))^2}] \\
&= \mathbb{E}_{x}[2\sum_{k=1}^{K}\sqrt{\frac{1}{n^2}\sum_{i=1}^{n}g_j(x_i;w_k)^2}] \\
&\leq \mathbb{E}_{x}[2\sum_{k=1}^{K}\sqrt{\frac{1}{n}}] \\
&= \frac{2K}{\sqrt{n}}.
\end{aligned}
$$

The desired result follows by applying Theorem 3 to $\hat{\mathscr{F}}_\theta, \hat{\mathscr{F}}_\theta(1), \cdots, \hat{\mathscr{F}}_\theta(M)$ and $\mathcal{D}$. $\qquad\square$

Next, we give the definition of semi-empirical risk. The term "semi-" implies that it is empirical with respect to the training set but not the smoothing operation.

**Definition 5.** *(Semi-empirical risk). For a surrogate loss function $l(\cdot,\cdot) : \mathbb{R}^M \times \mathbb{R}^M \to \mathbb{R}$ and training set $\{(x_i, y_i)\}_{i=1}^{n}$, the semi-empirical risk of $\phi(x) = \sum_{k=1}^{K}w_kg(x;\theta_k) \in \hat{\mathscr{F}}_\theta$ are defined as*

$$
\mathcal{R}_{se}[\phi] = \frac{1}{n}\sum_{i=1}^{n}l(\sum_{k=1}^{K}w_kg(x_i;\theta_k), y_i). \tag{21}
$$

We can use Lemma 4 and 10 to prove the following result:

**Theorem 4.** *Suppose for all $y \in \mathcal{Y}$, $l(\cdot, y)$ is a Lipschitz function with constant L and is uniformly bounded. Fix $\phi \in \mathscr{F}_p$, then for any $\eta > 0$, with probability at least $1 - \eta$ over the training dataset $\{(x_i, y_i)\}_{i=1}^{n}$ drawn i.i.d. from $\mathcal{D}$ and the parameters $\theta_1, ..., \theta_K$ drawn i.i.d. from $p$, the semi-empirical risk minimizer $\hat{\phi}$ over $\hat{\mathscr{F}}_\theta$ satisfies*

$$
\mathcal{R}[\hat{\phi}] - \mathcal{R}[\phi] < \frac{4\beta LMK + 4\sqrt{2\log\frac{8}{\eta}}}{\sqrt{n}} + 2L\sqrt{\|\phi\|_p}\sqrt[4]{\frac{M}{K}}(1 + \sqrt{2\log\frac{2}{\eta}})^{\frac{1}{2}},
$$

*where $\beta$ is a constant.*

*Proof.* Let $\phi^*$ be the minimizer of $\mathcal{R}$ over $\hat{\mathscr{F}}_\theta$. Combine Lemma 4 and 10, we derive that, with probability at least $1 - 2\delta$ over the training dataset and the choice of the parameters $\theta_1, ..., \theta_K$,

$$
\begin{aligned}
\mathcal{R}[\hat{\phi}] - \mathcal{R}[\phi] &= (\mathcal{R}[\hat{\phi}] - \mathcal{R}_{se}[\hat{\phi}]) + (\mathcal{R}_{se}[\hat{\phi}] - \mathcal{R}_{se}[\phi^*]) + (\mathcal{R}_{se}[\phi^*] - \mathcal{R}[\phi^*]) + (\mathcal{R}[\phi^*] - \mathcal{R}[\phi]) \\
&< \frac{2\beta LMK + 2\sqrt{2\log\frac{4}{\eta}}}{\sqrt{n}} + 0 + \frac{2\beta LMK + 2\sqrt{2\log\frac{4}{\eta}}}{\sqrt{n}} + 2L\sqrt{\|\phi\|_p}\sqrt[4]{\frac{M}{K}}(1 + \sqrt{2\log\frac{1}{\eta}})^{\frac{1}{2}} \\
&= \frac{4\beta LMK + 4\sqrt{2\log\frac{4}{\eta}}}{\sqrt{n}} + 2L\sqrt{\|\phi\|_p}\sqrt[4]{\frac{M}{K}}(1 + \sqrt{2\log\frac{1}{\eta}})^{\frac{1}{2}}.
\end{aligned}
$$

$\square$

**Lemma 11.** *Let $\mu$ be a probability distribution on $\Delta$. For any $\eta > 0$, with probability at least $1 - \eta$ over $x_1, ..., x_s$ drawn i.i.d. from $\mu$, it holds that*

$$
\|\frac{1}{s}\sum_{i=1}^{s} x_i - \mathbb{E}_{x\sim\mu}[x]\|_2 \leq \frac{1}{\sqrt{s}}(1 + \sqrt{2\log\frac{1}{\eta}}) \tag{22}
$$

*Proof.* Define $u(x_1, \cdots, x_s) = \|\frac{1}{s}\sum_{i=1}^{s} x_i - \mathbb{E}[x]\|_2$. By using Jensen's inequality, we have

$$
\mathbb{E}[u(x)] \leq \sqrt{\mathbb{E}[u^2(x)]} = \sqrt{\mathbb{E}[\|\frac{1}{s}\sum_{i=1}^{s} x_i - \mathbb{E}[x]\|_2^2]} = \sqrt{\frac{1}{s}(\mathbb{E}\|x\|_2^2 - \|\mathbb{E}[x]\|_2^2)} \leq \frac{1}{\sqrt{s}}.
$$

Next, for $x_1, \cdots, x_M$ and $\tilde{x}_k$, we have

$$
\begin{aligned}
&|u(x_1, \cdots, x_s) - u(x_1, \cdots, \tilde{x}_k, \cdots, x_s)| \\
&= |\|\frac{1}{s}\sum_{i=1}^{s} x_i - \mathbb{E}[x]\|_2 - \|\frac{1}{s}(\sum_{i=1,i\neq k}^{s} x_i + \tilde{x}_k) - \mathbb{E}[x]\|_2| \\
&\leq \|\frac{1}{s}\sum_{i=1}^{s} x_i - \frac{1}{s}(\sum_{i=1,i\neq k}^{s} x_i + \tilde{x}_k)\|_2 \\
&= \frac{\|x_k - \tilde{x}_k\|_2}{s} \\
&\leq \frac{2}{s}.
\end{aligned}
$$

Now we can use McDiarmid's inequality to bound $u(x)$, which gives

$$
\mathbb{P}[u(x) - \frac{1}{\sqrt{s}} \geq \varepsilon] \leq \mathbb{P}[u(x) - \mathbb{E}u(x) \geq \varepsilon] \leq \exp(-\frac{s\varepsilon^2}{2}). \tag{23}
$$

The result follows by setting $\eta$ to the right hand side and solving $\varepsilon$. $\square$

Now we are ready to prove Theorem 2.

*Proof of Theorem 2.* Let $\phi_0 \in \mathscr{F}_p$ such that $\mathcal{R}[\phi_0] < \inf_{\phi\in\mathscr{F}_p} \mathcal{R}[\phi] + \frac{\varepsilon}{4}$. By Lemma 11, with probability at least $1 - \frac{\eta}{3}$,

$$
\|\frac{1}{s}\sum_{j=1}^{s} f(x_i + \delta_{ijk}; \theta_k) - g(x_i; \theta_k)\|_2 \leq \frac{1 + \sqrt{2\log\frac{3Kn}{\eta}}}{\sqrt{s}}, 1 \leq i \leq n, 1 \leq k \leq K, \tag{24}
$$

hold simultaneously. So with probability at least $1 - \frac{\eta}{3}$, for every $\phi = \sum_{k=1}^{K} w_k g(x; \theta_k) \in \hat{\mathscr{F}}_\theta$, it holds that

$$
\begin{aligned}
|\mathcal{R}_{emp}[\phi] - \mathcal{R}_{se}[\phi]| &= |\frac{1}{n} \sum_{i=1}^{n} [l(\sum_{k=1}^{K} w_k [\frac{1}{s} \sum_{j=1}^{s} f(x_i + \delta_{ijk}; \theta_k)], y_i) - l(\sum_{k=1}^{K} w_k g(x_i; \theta_k), y_i)]| \\
&\leq \frac{L}{n} \sum_{i=1}^{n} \| \sum_{k=1}^{K} w_k [\frac{1}{s} \sum_{j=1}^{s} f(x_i + \delta_{ijk}; \theta_k) - g(x_i; \theta_k)] \|_2 \\
&\leq \frac{L}{n} \sum_{i=1}^{n} \sum_{k=1}^{K} w_k \| \frac{1}{s} \sum_{j=1}^{s} f(x_i + \delta_{ijk}; \theta_k) - g(x_i; \theta_k) \|_2 \\
&\leq \frac{L}{n} \sum_{i=1}^{n} \sum_{k=1}^{K} w_k \frac{1 + \sqrt{2 \log \frac{3Kn}{\eta}}}{\sqrt{s}} \\
&= \frac{L(1 + \sqrt{2 \log \frac{3Kn}{\eta}})}{\sqrt{s}} \triangleq \varepsilon_1.
\end{aligned}
$$

By Lemma 10, with probability at least $1 - \frac{\eta}{3}$, for every $\phi \in \hat{\mathscr{F}}_\theta$, it holds that

$$
|\mathcal{R}_{se}[\phi] - \mathcal{R}[\phi]| \leq \frac{2\beta L M K}{\sqrt{n}} + \sqrt{\frac{8 \log \frac{12}{\eta}}{n}} \triangleq \varepsilon_2.
$$

Let $\phi^*$ be the minimizer of $\mathcal{R}$ over $\hat{\mathscr{F}}_\theta$. By Lemma 4, with probability at least $1 - \frac{\delta}{3}$, for $K \geq M \|\phi_0\|_p^2 (1 + \sqrt{2 \log \frac{3}{\eta}})^2$,

$$
\mathcal{R}[\phi^*] - \mathcal{R}[\phi_0] < 2L\sqrt{\|\phi\|_p} \sqrt[4]{\frac{M}{K}} (1 + \sqrt{2 \log \frac{3}{\eta}})^{\frac{1}{2}} \triangleq \varepsilon_3.
$$

So with probability at least $1 - \eta$, it holds that

$$
\begin{aligned}
\mathcal{R}[\hat{\phi}] - \inf_{\phi \in \mathscr{F}_p} \mathcal{R}[\phi] &= (\mathcal{R}[\hat{\phi}] - \mathcal{R}_{se}[\hat{\phi}]) + (\mathcal{R}_{se}[\hat{\phi}] - \mathcal{R}_{emp}[\hat{\phi}]) + (\mathcal{R}_{emp}[\hat{\phi}] - \mathcal{R}_{emp}[\phi^*]) \\
&\quad + (\mathcal{R}_{emp}[\phi^*] - \mathcal{R}_{se}[\phi^*]) + (\mathcal{R}_{se}[\phi^*] - \mathcal{R}[\phi^*]) + (\mathcal{R}[\phi^*] - \mathcal{R}[\phi_0]) + (\mathcal{R}[\phi_0] - \inf_{\phi \in \mathscr{F}_p} \mathcal{R}[\phi]) \\
&< \varepsilon_2 + \varepsilon_1 + 0 + \varepsilon_1 + \varepsilon_2 + \varepsilon_3 + \frac{\varepsilon}{4} \\
&= 2\varepsilon_1 + 2\varepsilon_2 + \varepsilon_3 + \frac{\varepsilon}{4}.
\end{aligned}
$$

When $K > \frac{256 L^4 \|\phi_0\|_p^2 M (1 + \sqrt{2 \log \frac{1}{\eta}})^2}{\varepsilon^4}$, $n > \frac{64(2\beta L M K + \sqrt{8 \log \frac{12}{\eta}})^2}{\varepsilon^2}$, $s > \frac{64 L^2 (1 + \sqrt{2 \log \frac{3Kn}{\eta}})^2}{\varepsilon^2}$, we have

$$
\mathcal{R}[\hat{\phi}] - \inf_{\phi \in \mathscr{F}_p} \mathcal{R}[\phi] < \varepsilon. \tag{25}
$$

If $\mathcal{R}[\phi_0] = \inf_{\phi \in \mathscr{F}_p} \mathcal{R}[\phi]$, which means $\|\phi_0\|_p$ is independent of $\varepsilon$, $K = \Omega(\frac{1}{\varepsilon^4})$. $\qquad\square$

# B ALGORITHMS

## B.1 ADAPTIVE PREDICTION ALGORITHM

The entire adaptive prediction algorithm is shown in Algorithm 1. It is modified from Inoue (2019) to generalize to weighted ensembles. The exit condition is the weighted version of the confidence-level-based early-exit condition in Inoue (2019). The algorithm accelerates the evaluation of the ensemble function $f$ and the smoothed operation $g(x) = \mathbb{E}_\delta[f(x + \delta)]$ remains the same, so it does not affect the Monte Carlo estimation and certification procedure of smoothed classifiers.

---

**Algorithm 1** Adaptive prediction for weighted ensembling

---

1: **Input:** Ensembling weight $w \in \mathbb{R}^K$, candidate model parameters $\theta \in \Theta^K$, significance level $\alpha$, threshold $T$, data point $x$
2: Compute $z = \Phi^{-1}(1 - \frac{\alpha}{2})$
3: Set $\pi$ as the permutation of indices that sorts $w$ in descending order and $i \leftarrow 0$
4: **repeat**
5:     Set $i \leftarrow i + 1$
6:     Compute the $w_{\pi_i}$-th local prediction $p_{\pi_i} \leftarrow (p_{\pi_i,1}, \cdots, p_{\pi_i,M}) \in \Delta$
7:     Compute $\hat{p}_{i,k} \leftarrow \frac{\sum_{j=1}^{i} w_{\pi_j} p_{\pi_j,k}}{\sum_{j=1}^{i} w_{\pi_j}}$ for $k = 1, 2, \cdots M$
8:     Compute $k_i \leftarrow \arg\max_k \hat{p}_{i,k}$
9: **until** $\hat{p}_{1,k_1} > T$ or $\hat{p}_{i,k_i} > \frac{1}{2} + z \frac{\sqrt{\sum_{j=1}^{i} w_{\pi_j}^2}}{\sum_{j=1}^{i} w_{\pi_j}} \sqrt{\frac{\sum_{j=1}^{i} w_{\pi_j}(p_{\pi_j,k_i} - \hat{p}_{i,k_i})^2}{\sum_{j=1}^{i} w_{\pi_j}}}, i > 1$ or $i = K$
10: **return** $k_i$ and $\hat{p}_{i,k}, k = 1, 2, \cdots M$

---

### B.2 DETAILED SWEEN ALGORITHM

---

**Algorithm 2** SWEEN

---

1: **Input:** Training set $\hat{p}_{\text{train}}$, evaluation set $\hat{p}_{\text{eval}}$, Ensembling weight $w \in \mathbb{R}^K$, candidate model parameters $\theta = \{\theta_1, ..., \theta_K\} \in \Theta^K$.
2: Initialize $\theta_1, ..., \theta_K, w$
3: **for** $i = 1$ to $K$ **do**
4:     Train candidate models $\theta_i$ using $\hat{p}_{\text{train}}$.
5: **end for**
6: Construct SWEEN model $g_{\text{sween}}(\cdot; \theta, w) = \sum_{k=1}^{K} w_k g(\cdot; \theta_k)$
7: Train $w$ using $\hat{p}_{\text{eval}}$
8: **return** $w$ and $\theta_1, ..., \theta_K$

---

## C SUPPLEMENTARY MATERIAL FOR EXPERIMENTS

### C.1 DETAILED SETTINGS AND HYPER-PARAMETERS

We perform all experiments on CIFAR-10 and SVHN with a single GeForce GTX 1080 Ti GPU. For the experiments on ImageNet, we use eight V100 GPUs.

For training the SWEEN models on CIFAR-10 and SVHN, we divide the training set into two parts, one for training candidate models, and the other for solving weights. We employ 2,000 images for solving weights on CIFAR-10 and 3000 images for solving weights on SVHN. For ImageNet, we use the whole training set to train candidate models and 1/1000 of the training set to solve weights.

For Gaussian data augmentation training, all the models are trained for 400 epochs using SGD on CIFAR-10 and SVHN. The models on ImageNet are trained for 90 epochs. The learning rate is initialized set as 0.01, and decayed by 0.1 at the 150th/300th epoch.

For MACER training, we use the same hyper-parameters as Zhai et al. (2020), i.e., we use $k = 16, \beta = 16.0, \gamma = 8.0$, and we use $\lambda = 12.0$ for $\sigma = 0.25$ and $\lambda = 4.0$ for $\sigma = 0.50$. We train the models for 440 epochs, the learning rate is initialized set as 0.01, and decayed by 0.1 at the 200th/400th epoch.

### C.2 RESULTS ON SVHN

To further evaluate our method, we also experiment on SVHN. The results in Table 5 show that SWEEN models outperform the upper envelopes of their corresponding candidate models as well. Figure 2 plots the results on CIFAR-10 and SVHN for comparison.

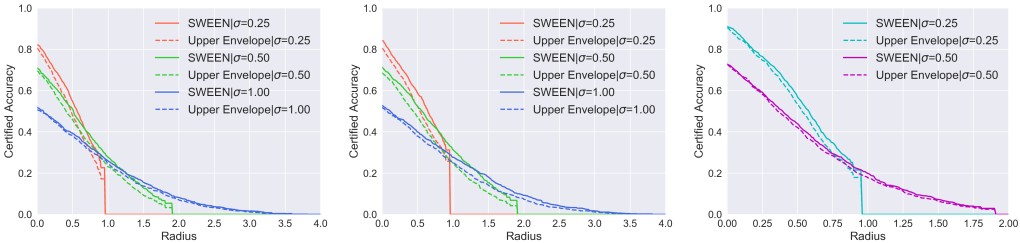

Figure 2: Comparing SWEEN models to the upper envelopes of their corresponding candidate models. All models are trained via the standard training. (**Left**) The SWEEN-3 model on CIFAR-10. (**Middle**) The SWEEN-7 model on CIFAR-10. (**Right**) The SWEEN-7 model on SVHN.

Table 5: Certified accuracy (%) and ACR on SVHN. All models are trained via standard training. UE stands for the upper envelope of candidate models.

| $\sigma$ | Model | 0.00 | 0.25 | 0.5 | 0.75 | 1.00 | 1.25 | 1.50 | 1.75 | 2 | ACR |
|---|---|---|---|---|---|---|---|---|---|---|---|
| 0.25 | UE$_{SWEEN-7}$ | 90.3 | 74.6 | 53.4 | 29.5 | 0 | 0 | 0 | 0 | 0 | 0.517 |
|  | SWEEN-7 | 91.0 | 76.4 | 56.8 | 34.7 | 0 | 0 | 0 | 0 | 0 | 0.547 |
| 0.50 | UE$_{SWEEN-7}$ | 72.7 | 58.1 | 42.1 | 28.4 | 17.7 | 11.0 | 6.5 | 3.1 | 0 | 0.498 |
|  | SWEEN-7 | 72.8 | 59.2 | 43.9 | 29.4 | 19.8 | 11.9 | 7.3 | 3.7 | 0 | 0.524 |

## C.3 SWEEN MODELS USING CANDIDATES WITH IDENTICAL ARCHITECTURES

The SWEEN-3 and SWEEN-7 models are all using candidate models with diverse architectures. For a more comprehensive result, we also experiment with SWEEN models using candidates with identical architectures. For $\sigma \in \{0.25, 0.5, 1.0\}$, We train 8 ResNet-110 models using different random seeds on CIFAR-10 via the standard training. We then use these models to ensemble SWEEN models.

The results are shown in Table 6 and Figure 3. We can see that SWEEN is still effective in this scenario and significantly boosts the performance compared to candidate models.

We also run experiments on ImageNet using models with identical structure but with different random initialization. We train 3 ResNet-50 on ImageNet via the standard training to ensemble the SWEEN models. Table 7 shows the results. The improvement of SWEEN is substantial compared with the AVG and UE results.

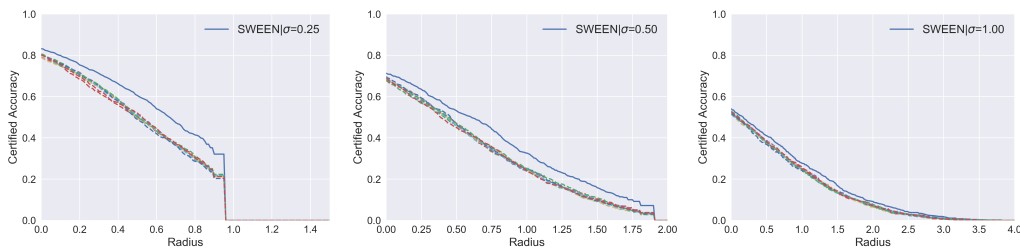

Figure 3: Radius-accuracy curves of SWEEN models and their candidate models. All candidate models are using the ResNet-110 architecture and trained via the standard training. (**Left**) $\sigma = 0.25$. (**Middle**) $\sigma = 0.50$. (**Right**) $\sigma = 1.00$.

Table 6: ACA (%) and ACR on CIFAR-10. All candidate models are ResNet-110s trained via the standard training. UE stands for the upper envelope, which shows the largest ACA and ACR among the candidate models. AVG stands for the average ACA or ACR of candidate models.

| $\sigma$ | Model | 0.00 | 0.25 | 0.5 | 0.75 | 1.00 | 1.25 | 1.50 | 1.75 | 2.00 | ACR |
|---|---|---|---|---|---|---|---|---|---|---|---|
| 0.25 | AVG | 79.9 | 67.2 | 50.3 | 33.5 | 0 | 0 | 0 | 0 | 0 | 0.491 |
| | UE | 80.4 | 68.3 | 52.0 | 34.6 | 0 | 0 | 0 | 0 | 0 | 0.496 |
| | SWEEN | 83.5 | 73.0 | 60.4 | 43.8 | 0 | 0 | 0 | 0 | 0 | **0.572** |
| 0.50 | AVG | 68.4 | 58.0 | 46.1 | 34.5 | 24.4 | 16.2 | 9.7 | 4.4 | 0 | 0.568 |
| | UE | 69.7 | 59.3 | 47.1 | 35.7 | 24.9 | 17.6 | 10.8 | 5.1 | 0 | 0.581 |
| | SWEEN | 71.3 | 63.3 | 53.1 | 44.0 | 32.6 | 22.9 | 15.8 | 9.2 | 0 | **0.691** |
| 0.50 | AVG | 51.9 | 45.0 | 37.8 | 30.9 | 24.7 | 18.7 | 13.5 | 9.9 | 7.1 | 0.558 |
| | UE | 53.0 | 46.0 | 38.6 | 31.8 | 25.5 | 19.3 | 14.1 | 10.5 | 7.7 | 0.566 |
| | SWEEN | 54.1 | 47.3 | 41.0 | 34.7 | 27.6 | 22.9 | 16.7 | 12.2 | 9.2 | **0.623** |

Table 7: ACA (%) and ACR on ImageNet. All candidate models are ResNet-50s trained via the standard training. The SWEEN model here contains 3 ResNet-50s. UE stands for the upper envelope, which shows the largest ACA and ACR among the candidate models. AVG stands for the average ACA or ACR of candidate models.

| $\sigma$ | Model | 0.00 | 0.25 | 0.5 | 0.75 | 1.00 | 1.25 | 1.50 | 1.75 | 2.00 | ACR |
|---|---|---|---|---|---|---|---|---|---|---|---|
| 0.50 | AVG | 56.8 | 52.1 | 46.1 | 42.3 | 37.5 | 32.8 | 28.1 | 21.6 | 0 | 0.723 |
| | UE | 57.2 | 52.4 | 46.4 | 42.4 | 37.8 | 33.0 | 28.2 | 21.8 | 0 | 0.727 |
| | SWEEN | 60.0 | 55.2 | 50.2 | 46.0 | 42.6 | 37.8 | 33.2 | 28.4 | 0 | **0.816** |

## C.4 ADAPTIVE PREDICTION ENSEMBLING

To alleviate the higher execution cost introduced by SWEEN, we apply the previously mentioned adaptive prediction algorithm to speed up the certification. Experiments are conducted on the SWEEN-7 models via the standard training on CIFAR-10 and the results are summarized in Table 8. It can be observed that the adaptive prediction successfully reduce the number of evaluations. However, the performance of the adaptive prediction models is only slightly worse than their vanilla counterparts.

Table 8: ACA (%) and ACR on CIFAR-10. All models are trained via the standard training. * means the upper envelope of candidate models.

| $\sigma$ | Model | 0.00 | 0.25 | 0.5 | 0.75 | 1.00 | 1.25 | 1.50 | 1.75 | ACR | #evals/img |
|---|---|---|---|---|---|---|---|---|---|---|---|
| 0.25 | Normal | 84.2 | 72.0 | 58.7 | 43.0 | 0 | 0 | 0 | 0 | 0.560 | 700,700 |
| | Adaptive | 84.3 | 71.5 | 57.6 | 41.2 | 0 | 0 | 0 | 0 | 0.549 | 283,727 |
| 0.50 | Normal | 71.2 | 63.0 | 52.2 | 41.9 | 31.2 | 22.9 | 15.3 | 8.3 | 0.678 | 700,700 |
| | Adaptive | 70.9 | 62.8 | 52.3 | 41.6 | 31.1 | 22.8 | 14.5 | 7.8 | 0.672 | 382,426 |

## C.5 THE SCATTER PLOT OF THE CERTIFIED ACCURACY

This section plots the scatter diagram of the certified accuracy of test data points for SWEEN-7 versus ResNet-110 in Figure 4. Most of the points lie under the line $y = x$, implying that SWEEN-7 performs superior to ResNet-110.

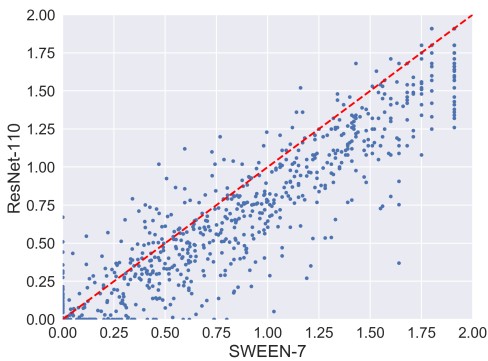

Figure 4: The scatter diagram of the certified accuracy of test data points for SWEEN-7 versus ResNet-110. $\sigma = 0.50$.

## C.6 SWEEN VERSUS ADVERSARIAL ATTACKS

We further investigate the performance of SWEEN models versus AutoAttack (Croce & Hein, 2020), which is an ensemble of four diverse attacks to reliably evaluate robustness. Similar to Salman et al. (2019a), we used 128 samples to estimate the smoothed classifier. We share the results below in Table 9. It can be seen that SWEEN can improve the empirical robustness as well.

Table 9: Certified accuracy and empirical accuracy versus AutoAttack on CIFAR-10. All candidate models are trained via the standard training.

| $\sigma$ | Model | 0.00 | 0.25 | 0.5 | 0.75 | 1.00 |
|---|---|---|---|---|---|---|
| | SWEEN-3 (certified) | 70.9 | 61.4 | 50.8 | 38.3 | 27.7 |
| | SWEEN-3 (AA) | **75.0** | **67.0** | **58.9** | **48.4** | **39.9** |
| 0.50 | ResNet-20 (AA) | 72.5 | 64.7 | 55.4 | 46.2 | 37.0 |
| | ResNet-26 (AA) | 74.5 | 65.4 | 57.3 | 46.3 | 35.7 |
| | ResNet-32 (AA) | 73.8 | 64.5 | 55.3 | 45.0 | 35.3 |

## C.7 MACER TRAINING ON CIFAR-10

In this section, we plot the radius-accuracy curves for SWEEN models with candidate models trained by MACER on CIFAR-10 in Figure 5.

## D FIGURES

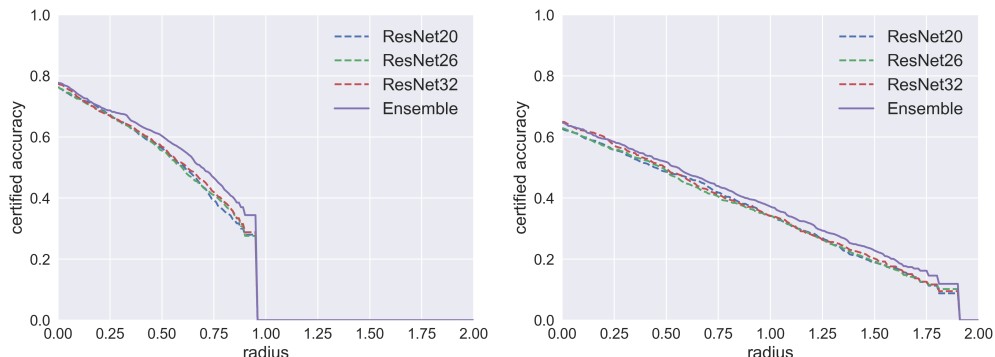

Figure 5: Radius-accuracy curves for SWEEN models with candidate models trained by MACER on CIFAR-10. (**Left**) $\sigma = 0.25$. (**Right**) $\sigma = 0.50$.

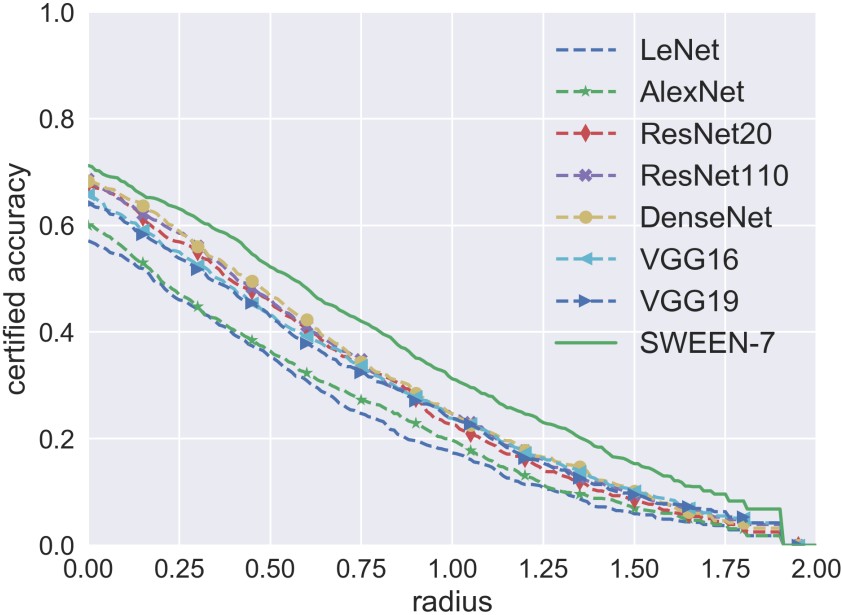

Figure 6: Radius-accuracy curves w.r.t. the SWEEN-7 model and all its candidate models under $\sigma = 0.50$. All models are trained via the standard training.

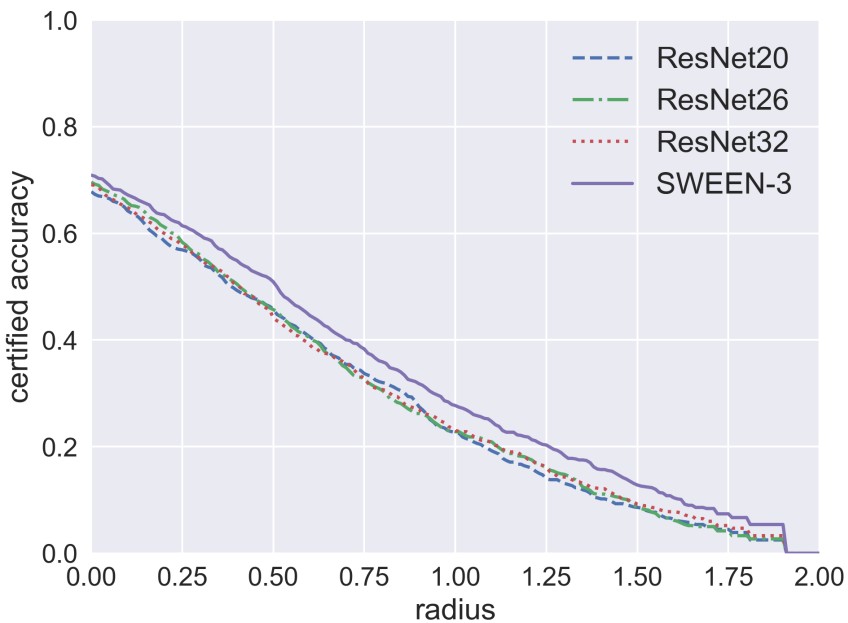

Figure 7: Radius-accuracy curves w.r.t. the SWEEN-3 model and all its candidate models under $\sigma = 0.50$. All models are trained via the standard training.

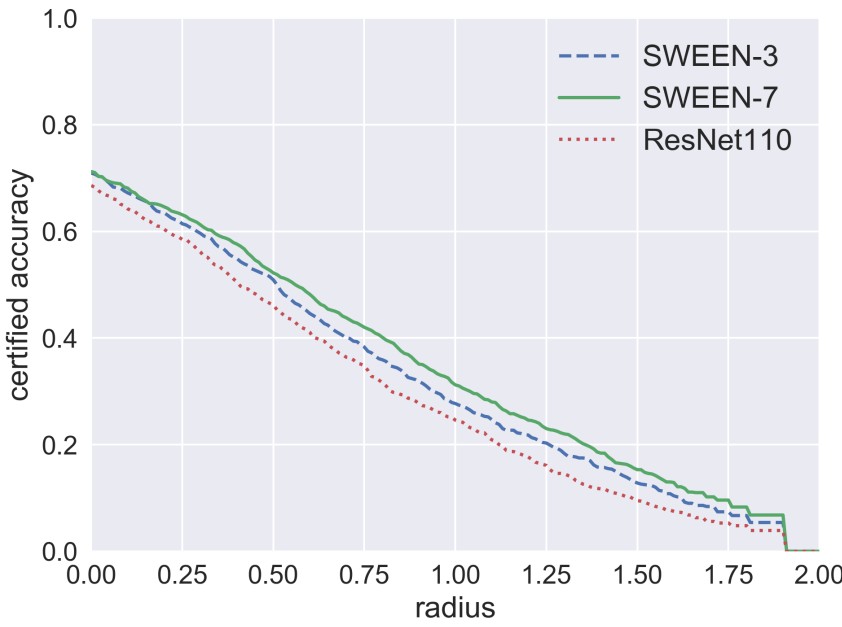

Figure 8: Radius-accuracy curves w.r.t. the SWEEN-7 model, the SWEEN-3 model and the ResNet-110 under $\sigma = 0.50$. All models are trained via the standard training.

