# OpenReview forum: "Enhancing Certified Robustness of Smoothed Classifiers via Weighted Model Ensembling"
_ICLR.cc/2021/Conference — Reject_

### Official Review · AnonReviewer1 · 2020-10-27
**Official Blind Review #1**

**Rating:** 5
**Confidence:** 4

**Review:**

The paper claims that an ensemble of multiple classifiers can achieve better certified robustness than a single model with comparable cost in terms of the total # parameters. It also proposes to optimize the ensemble weights for maximizing the robustness, and an adaptive prediction scheme to reduce extra # inferences from the ensemble computation required for the certification. Experimental results show that their proposed ensemble models (which are smaller than ResNet-110 in total) achieves better ACR than standard training with ResNet-110, and comparable ACR when applied to MACER.

Overall, the paper is clearly written and easy-to-follow. However, I generally feel both the theoretical and empirical results of the paper are somewhat weak to meet the ICLR bar for the current submission:
- It seems to me that Lemma 1 and Theorem 2 are somewhat straightforward (or incremental) implications of the (well-known) generalization theory (e.g., [1]), and I could not find a significant reason why they should be presented in the main text. The paper could highlight why those results are surprising compared to existing generalization bounds.
- I think Algorithm 1, on the other hand, should be rather justified more rigorously instead of SWEEN itself: "could Algorithm 1 really recover the MC estimate?", or "is there a theoretical guarantee that this algorithm is more efficient than MC?", just to name a few. Also, the paper should provide more detailed explanation and rationale on Algorithm 1.
- The empirical results are not fully convincing to show that model ensemble reliably improve robustness: the paper only present two specific configurations of ensemble, namely SWEEN-3 and SWEEN-7, but there should be much more combinations to verify the effectiveness of ensemble. Also, I feel the results on MACER ensemble (Table 2) are too marginal.

[1] Kawaguchi et al., Generalization in Deep Learning, 2017.

Minor comments
- p4, "the volume of the certified region are more comprehensive ... ": then why the volume measure was not considered for evaluation in the experimental results?
- p5, "... will require 10,010,000 local evaluations ...": it would be a bit confusing for some readers to figure out why it requires additional 10,000 inferences in the count. I guess it is from a rough prediction cost in the CERTIFY algorithm, but I could not find any related explanation of this in the paper.
- Could SWEEN further improve ResNet-110 if one consider ResNet-110 x 3 ensemble model?
- Why SWEEN-7 is not considered in Table 2?

---

> ### Author Response · Authors · 2020-11-18
> **Author Response (Part 1)**
>
> We thank the reviewer for thoroughly reading our paper and carefully checking our theoretical analysis and experiments. Here are our responses to your questions and concerns:
>
> [Regarding the novelty of Lemma 1 and Theorem 2]
>
> We respectfully disagree with the comment that Lemma 1 and Theorem 2 are incremental implications of the well-known generalization theory. We would like to restate our theoretical part and then show the difference between the generalization theory and our results.
>
> (a)	Lemma 1 gives the result that ensembling in the space can achieve optimal $\\gamma$-robustness in $\\mathscr F_p$, which is independent of the generalization theory. The essence of Lemma 1 is to extend the optimal risk in the discrete ensembling space $\hat{\mathscr{F}_{\theta}}$ to the more expressive continuous space $\mathscr{F}_p$, which is exactly the benefits of ensembling.
>
> (b) Theorem 2 states that the SWEEN model can be easily trained to achieve near-optimal robustness. We decompose the robustness gap between empirically optimized SWEEN $\\hat{\\phi}$ and the optimal robustness, $\\mathcal{R}[\\hat{\\phi}] - \\inf \\limits_{\\phi \\in \\mathscr{F}_p} \\mathcal{R}[\\phi]$, into several parts:
>
> $$
> \begin{eqnarray*}
> \\mathcal{R}[\\hat{\\phi}]-\\inf_{\\phi\\in\\mathscr{F}_p}\\mathcal{R}[\\phi] &=& (\\mathcal{R}[\\hat{\\phi}]-\\mathcal{R}_\{se}[\\hat{\\phi}]) + (\\mathcal{R}_\{se}[\\hat{\\phi}]-\\mathcal{R}_\{emp}[\\hat{\phi}])+(\\mathcal{R}_\{emp}[\\hat{\\phi}]-\\mathcal{R}_\{emp}[\\phi^*])+\\\\
> &&(\\mathcal{R}_\{emp}[\phi^*]-\\mathcal{R}_\{se}[\phi^*])+(\\mathcal{R}_\{se}[\phi^*] - \\mathcal{R}[\phi^*])+(\\mathcal{R}[\phi^*]-\\mathcal{R}[\phi_0])+(\\mathcal{R}[\phi_0]-\\inf_\{\phi\in\\mathscr{F}_p}\\mathcal{R}[\phi]).
> \end{eqnarray*}
> $$
>
> Only the first and fifth items are bounded by the standard generalization theory (Lemma 7). The sixth part, the gap between the optimal discrete ensemble and the optimal $\phi_0$ in the continuos space $\mathscr{F}_p$, can only be bounded by Lemma 4 (the core of Lemma 1). What's more, the second and fourth items are bounded by Jensen's inequality and McDiarmid's inequality (Lemma 11); the third and the last items are bounded by the definition of the minimizer. Using the generalization bound cannot go through this process.
>
> Another way to use the generalization theory is to apply it to both $\phi_0$ and $\hat{\phi}$. Ignoring the details on bounding two Rademacher terms, we still face the gap between $\mathcal{R}_\{se}[\hat{\phi}]$ and $\mathcal{R}_\{se}[\phi_0]$ (or $\mathcal{R}_\{emp}[\hat{\phi}]$ and $\mathcal{R}_\{emp}[\phi_0]$). Bounding either term requires comparing the optimal discrete ensemble and the optimal $\phi_0$ in the continuous space $\mathscr{F}_p$, and we still need Lemma 1.
>
> [Regarding Algorithm 1]
>
> In our SWEEN function, using Monte Carlo estimation is independent of using Algorithm 1. According to Eqn(2) in Section 3, Monte Carlo is used to smooth the base function $f$ to $g$ while SWEEN proposes to use ensembling to better learn a base function $f_{ens}$. That is, Algorithm 1 is used to speed up the ensembling and replace $f_{ens}$ with $f_{ens}^{alg1}$. $f_{ens}^{alg1}$ will then be smoothed to provide certified robustness. Thus, Algorithm 1 will not affect the Monte Carlo estimation and certification procedure of smoothed classifiers.
>
> Nonetheless, we agree that there are questions related to Algorithm 1 that require further thinking. The stopping criterion in Algorithm 1 or the original algorithm in [Inoue 2019] is inspired by the confidence interval. How to theoretically analyze the gap introduced by algorithm 1 and how to balance the tradeoff between the speed-up and the gap are all meaningful questions for ensembling. But we think they are out of the scope of this paper.
>
> [Regarding the empirical results]
>
> We have listed extra results on SWEEN using identical architectures (ResNet-110s)  in Appendix B.3. The most commonly used architecture in adversarial robustness is the ResNet. Besides ensembling ResNets with different depths, we also include AlexNet, DenseNet, and VGG. We think that SWEEN-3 (limited number of candidate models in ResNets), SWEEN-7 (adequate candidate models with diverse architecture), and SWEEN with identical architectures are representative enough to show the effectiveness of SWEEN in different scenarios.
>
> We agree that if only the ACR is considered, ensembling MACER models provide marginal improvement. However, the training time is decreased, the number of parameters is lowered, and the number of FLOPs is smaller. We think when comparing results, not only the performance but the energy consumption should be considered. Thus, the improvement is beyond marginal.

---

> > ### Author Response · Authors · 2020-11-18
> > **Author Response (Part 2)**
> >
> > [Regarding other questions]
> >
> > - We did not present the results with respect to the volume of the certified region because prior works did not use it, and we cannot provide a fair comparison.
> > - We are sorry for the confusion. The additional inferences come from the prediction cost. We have made it clearer.
> > - Yes. As stated above, we provided results using SWEEN consisting of 8 ResNet-110s. Please see Appendix B.3. The ResNet-110 x 3 case should be similar.
> > - We found that MACER performs poorly in training models that are not ResNets. MACER may not converge on VGG models. Thus, we only present results of SWEEN-3, which only have ResNets.
> >
> > We sincerely hope that the reviewer can check our response and let us know whether all your questions and concerns have been addressed. It would be great if you could upgrade your rating if you are satisfied with our response. Thank you for your re-consideration.

---

> > > ### Comment · AnonReviewer1 · 2020-11-23
> > > **Additional question on Lemma 1 and Theorem 2**
> > >
> > > Thanks for your detailed clarification, and I can see that there can be more technical details than the standard generalization theory to connect itself to the certified robustness.
> > >
> > > It is still unclear to me, however, that why these theorems should be highlighted in the main text - more specifically, why the continuous mixture $\mathcal{F}_{p}$ should be considered to explain the effectiveness of SWEEN? I agree that it is super-natural to say SWEEN can arbitrary approximate its continuous extension, but does it really gives some insights on why SWEEN-7 should be better than standard ResNet-110? As both Lemma 1 and Theorem 2 assumes a sufficiently large $K$ (the # models), I still do not think their experimental results could be actually connected to its theoretical claims in any aspect.

---

> > > > ### Author Response · Authors · 2020-11-23
> > > > **Response to AnonReviewer1**
> > > >
> > > > We agree that our theoretical results cannot demonstrate that SWEEN-7 performs strictly better than ResNet-110. Actually, such a claim is not correct. For example, it is impossible to show the superiority of ensemble models if one of the candidate models achieves the possible optimal performance. The best result is that SWEEN-7 performs not worse than ResNet-110, which is trivial since SWEEN-7 contains ResNet-110. However, we highlight the significance of our theoretical results in supporting SWEEN:
> > > >
> > > > - $\mathscr{F}_p$ is more expressive than one single model $f(\cdot ; \theta)$ with $\theta$ in a fixed $\Theta$, and training a SWEEN model consisting of small candidate models is much easier than training a single large model. Thus, SWEEN can be beneficial. It is supported by the empirical evidence that SWEEN-3 (ResNet-20, 26, and 32) performs better than each candidate model and even ResNet-110.
> > > >
> > > > - We acknowledge the gap between Theorem 2 and our experimental settings. However, our theoretical results can be considered a supplement to our empirical evidence. Lemma 1 and Theorem 2 are used to highlight the desired properties, e.g., ensembling generality and the accessibility of optimal $\gamma$-robustness with large $K$. In return, the experiments show that the certified robustness can be improved with only a few candidate models.
> > > >
> > > > In summary, we believe that our theoretical and experimental results support SWEEN from different perspectives, so we include both parts in the main text.

---

### Official Review · AnonReviewer2 · 2020-10-28
**An Interesting Work but Need Some Edits**

**Rating:** 6
**Confidence:** 4

**Review:**

This paper demonstrates the advantages of ensembles of smoothed classifiers to achieve better approximately provable robustness.
Theoreotically, it proves that under certain conditions, given enough amount of individual models, the function which the ensemble model represent can be archibitarily close to the optima within a function class. Empirically, the ensemble models have better performance in approximately provable robustness than an individual model of even more trainable parameters.

--Novelty and Contribution

Pros:

1. It is well known that adversarial robustness needs higher model capacity, so ensemble model can improve empirical robustness. Although not surprising, as far as I know, there is no prior work investigating ensemble in the context of randomized smoothing.

2. The experimental result shows ensemble models can really improve the robustness. The results on CIFAR10 and SVHN is comprehensive. The authors also show that adaptive prediction ensembling can accelerate the computation.

Cons:

1. There is a significant gap between theoretical analysis and experimental settings. Based on the condition above Equation (25), both the number of individual models and the number of training samples are trivially large. Theorem 2 should also emphasize the need of sufficient training samples.

2. One assumption of Theorem 2: $l(0, y) = 0$ might be inconsistent with the practice. The most popular loss function is cross-entropy loss, which does not meet this assumption. I notice that this assumption is first used in Lemma 8 to prove Theorem 3. Removing this condition might result in a negative term on RHS of Theorem 4. I think the author should discuss on this assumption at least.

3. It is better to have results on ImageNet or its downsampled version,  like [Salman 2019].

--Presentation

Pros:

Generally, this paper is well-written. The notation and formulation are easy to follow.

Cons:

1. The author should discuss more about the condition to finish the loop in Algorithm 1. This part is not well-justified. Especially, the authors should point out what is new in Algorithm 1 compared with [Inoue 2019].

2. It is better to make an algorithm box demonstrating the whole method.

--Summary

From my point of view, this is a borderline paper. It systematically study the ensemble of randomized smooth models and demonstrate interesting results. The idea and the algorithm itself is somehow not intriguing.

---

> ### Author Response · Authors · 2020-11-18
> **Author Response**
>
> We want to thank the reviewer for careful reading and valuable comments. Below we address the concerns mentioned in the review:
>
> [Regarding Theorem 2]
>
> - We acknowledge the gap between theoretical results and experimental settings. We have now added comments about the number of candidate models and training samples in Section 4.2.
>
> - We are grateful that the reviewer pointed out this! After checking our derivations, we found out that the assumption $l(0, y)=0$ is NOT needed in Theorem 2. We use Lemma 8 to prove Theorem 3, in whose proof Lemma 8 is applied to $c(\cdot,y)-c(0,y)$. With such shifting, Theorem 3 and results afterward, i.e., Lemma 10, Theorem 4, and Theorem 2, do not need the assumption anymore. We have modified the formulation of these results.
>
> [Regarding experiments]
>
> We are running experiments right now and will update the results as soon as possible.
>
>
>
> [Regarding Algorithms]
>
> - To make the content more compact, we have moved Algorithm 1 to Appendix B.1 and made more detailed comments about our adaptive prediction algorithm there.
>
> - We presented the algorithm for training a SWEEN model in Appendix B.2.
>
>
> We hope our responses can address your questions and concerns about our work. We would also be happy to answer any other questions you may have.

---

> > ### Comment · AnonReviewer2 · 2020-11-18
> > **Lemma 8**
> >
> > The assumption of Theorem 2 is still a bit confusing.
> >
> > $l(0, y) = 0$ is already an assumption of Lemma 8 ("c is such that ... passes through the origin and uniformly bounded"), in which the loss function is denoted $c$.
> > After checking Lemma 7, $\tilde{c}$ is not identical to $c$.
> > Actually, it is $c(\cdot, y) - c(0, y)$.
> >
> > By Lemma 6, we have $\exists \beta, R_n(\tilde{c} \circ \mathcal{F}) \leq \beta G_n(\tilde{c} \circ \mathcal{F})$.
> > However, LHS of the inequality in Lemma 8 is $G_n(c \circ \mathcal{F})$ instead of $G_n(\tilde{c} \circ \mathcal{F})$.
> > These two Lemmas are not connected without this assumption.
> >
> > Actually, Lemma 8 is taken from a reference and the proof is not provided in this paper.
> > The authors need to carefully check its assumption before applying it.

---

> > > ### Author Response · Authors · 2020-11-19
> > > **Response to AnonReviewer2**
> > >
> > > We are sorry we did not make it clear. Actually we apply Lemma 8 to $\\tilde\{c}$ instead of $c$ in our proof.
> > >
> > > In Theorem 3, $c(\\cdot, y)$ is a Lipschitz function with constant $L$ and is uniformly bounded. So $\\tilde\{c}(\\cdot, y) = c(\\cdot, y)-c(0, y)$ is a Lipschitz function with constant $L$ which passes through the origin and is uniformly bounded. Hence, we can apply Lemma 8 to $\\tilde\{c}(\\cdot, y) $ and derive
> > > $$
> > > G_n(\tilde c\circ\mathscr{F})\le 2L\sum_{i=1}^MG_n(\mathscr{F}_i).
> > > $$
> > >
> > > Combining Lemma 6, we have $R_n[\tilde{c}\circ \mathscr{F}]\le \beta G_n(\tilde c\circ\mathscr{F})\le 2\beta L\sum_{i=1}^MG_n(\mathscr{F}_i).$ Since $\beta$ can be any constant, $2$ can be absorbed into $\beta$ and that is Theorem 3.

---

> > > > ### Comment · AnonReviewer2 · 2020-11-23
> > > > **Thanks**
> > > >
> > > > Thank you for the clarification, I think this formula is crucial to understand why such condition can be dropped.

---

### Official Review · AnonReviewer3 · 2020-10-28
**An interesting finding for smoothing based certification**

**Rating:** 6
**Confidence:** 3

**Review:**

In this work, the authors study the effect of ensembling on randomized smoothing certification.  By learning diverse classifiers and then applying randomized smoothing the authors arrive at an ensemble of smoothed classifiers. Predictions are then made based on a linear combination of the models outputs which are weighted based on empirical risk and robustness of each model. The authors make an argument for the optimality of the proposed approach and show that it performs favorably compared to randomized smoothing certification of only a single model.

While I can confirm that the notion of certified robustness that the authors propose in (6) is sensible and sound in terms of probabilistic guarantees of robustness, it is not clear to me exactly how Theorem 2 is the consequence of a using the proposed SWEEN model and not a property which is generally true of ensembles. Using a linear combination of models based on their empirical risk seems to be something which one could do in general, just that in theorem 2 the authors are using a notion of risk which also includes the certified robustness of the models. Perhaps I have missed something here. Given some time, I will read further into the exact details in the supplementary which I was unable to do sufficiently during the initial review period.

The results that the authors show for their models are favorable and show increased robustness wrt the adversarial smoothing criterion. I wonder if the authors have any results on adversarial attacks (e.g. FGSM, PGD) as there is some notion that using a diverse set of models can lead to a more robust predictor wrt attacks [1] but then there is evidence that for deterministic ensembles this may not be the case [2].

[1] https://arxiv.org/abs/2002.04359

[2] https://nicholas.carlini.com/papers/2017_woot_ensembles.pdf


Post rebuttal response:

I thank the authors for their response to my review and I take their point that establishing theorem 2 points to a limitation of using smoothing with ensembles, but I think this point could be made much more clear in the main text. Following a reading of all of the other reviews and re-evaluating the paper I remain optimistic and slightly positive about the paper as I think it is an important and interesting research direction; however, I am not fully convinced to increase my score given that some of the empirical comparisons could be greatly strengthened to be more than marginal improvements over MACER trained networks. I think that any insights that arise during the process of strengthening the results would contribute to a better understanding of the method and a stronger paper.

---

> ### Author Response · Authors · 2020-11-18
> **Author Response**
>
> We thank the reviewer for the feedback to improve our paper.
>
> [Regarding theoretical results]
>
> We agree that Theorem 2 is not fundamentally different from its counterpart in general ensembling. However, we would like to highlight our efforts in overcoming the complications and technical difficulties introduced by certified robustness. The major differences are the robust loss and the Monte Carlo (MC) sampling used in smoothing the network. To address the $\gamma$-robustness index, we had to consider the property of $\mathscr{F}_p$ for technical details in Lemma 3. We also introduced a semi-empirical risk in Definition 5 for the MC estimation error in training.
>
> In a word, we propose Theorem 2 to justify the feasibility of SWEEN. The theoretical results are not meant to show the unique suitability of randomized smoothing to ensembles but rather highlight desirable properties (e.g., the attainability of optimal $\gamma$-robustness index).
>
> [Regarding SWEEN versus adversarial attacks]
>
> Thanks for the suggestion. We further investigated SWEEN models' performance versus AutoAttack[1], an ensemble of four diverse attacks to reliably evaluate robustness. Similar to [2], we used 128 samples to estimate the smoothed classifier. We show the results in the table below. It can be seen that SWEEN can improve the empirical robustness as well. We also added the results in Appendix C.6.
>
>
> | $\\sigma$ | Model                  | 0\.00             | 0\.25             | 0\.5              | 0\.75             | 1\.00             |
> |-----------|------------------------|-------------------|-------------------|-------------------|-------------------|-------------------|
> |           | SWEEN\-3 \(certified\)&nbsp;&nbsp; | 70\.9             | 61\.4             | 50\.8             | 38\.3             | 27\.7             |
> |           | SWEEN\-3 \(AA\)        | $\\textbf\{75\.0\}$&nbsp;&nbsp; | $\\textbf\{67\.0\}$&nbsp;&nbsp; | $\\textbf\{58\.9\}$&nbsp;&nbsp; | $\\textbf\{48\.4\}$&nbsp;&nbsp; | $\\textbf\{39\.9\}$&nbsp;&nbsp; |
> |   0.5 &nbsp;&nbsp;       | ResNet\-20 \(AA\)      | 72\.5             | 64\.7             | 55\.4             | 46\.2             | 37\.0             |
> |           | ResNet\-26 \(AA\)      | 74\.5             | 65\.4             | 57\.3             | 46\.3             | 35\.7             |
> |           | ResNet\-32 \(AA\)      | 73\.8             | 64\.5             | 55\.3             | 45\.0             | 35\.3             |
>
> We hope our responses fully address your concern about the paper. We would also be willing to answer any other questions you may have.
>
> References
>
> [1] Croce and Hein, Reliable evaluation of adversarial robustness with an ensemble of diverse parameter-free attacks. ICML 2020.
>
> [2] Salman et al.  A convex relax-ation barrier to tight robustness verification of neural networks. InNeurIPS 2019 : Thirty-thirdConference on Neural Information Processing Systems, pp. 9832–9842, 2019b

---

> > ### Comment · AnonReviewer4 · 2020-11-18
> > **AutoAttack results**
> >
> > I do have a small question concerning the empirical robustness results. Was AutoAttack setup to use "Expectation over Transformation (EoT)"?

---

> > > ### Author Response · Authors · 2020-11-18
> > > **Response to AnonReviewer4**
> > >
> > > Considering the computation cost, we used the default setting for AutoAttack. However, we think that 128 samples are enough to well approximate the gradient of smoothed classifiers as shown in [2]. Hence, using the EoT version of AutoAttack will not appreciably downgrade the performance.
> > >
> > > Nonetheless, we are happy to redo the experiment with the EoT version of AutoAttack and update the results in our next revision.

---

### Official Review · AnonReviewer4 · 2020-10-30
**Initial review**

**Rating:** 6
**Confidence:** 3

**Review:**

In this paper, the authors introduce SWEEN, an ensembling scheme for smoothed classifiers. This scheme uses pre-trained models and learns the weights used to sum the output predictions. During deployment, smoothed models rely on sampling a large number of input perturbations and ensembling exacerbates the computation burden. Hence, the authors also propose an adaptive scheme that automatically adjusts the number of samples.

Overall, the paper is well-written and provides solid evidence that ensembling can improve certified robustness.
1) In the experiments, the authors focus on SWEEN-3/7 with standard training and SWEEN-3 with MACER. Have the authors tried SWEEN-7 with MACER? (or simply to add the ResNet-110 to the SWEEN-3 ensemble). I am generally curious about the limits of the approach.
2) While the adaptive scheme is useful, it distracts the reader from the main message. I'd suggest moving it to the appendix and consider experiments on ImageNet.
3) The wording is slightly unclear on how exactly the ensemble weights are trained. From the supplementary material, I gathered that the weight are trained to minimize the cross-entropy loss on perturbed inputs (perturbed by Gaussian noise) - as done for "standard training". Is that correct?
4) It is unclear why the $\gamma$-robustness index needs to be introduced to demonstrate that SWEEN can be trained to near-optimal risk with a surrogate loss.

Details:
A) Fig. 1 is not color-blind friendly and difficult to read (small font).


---------
Update post-rebuttal: Thank you for addressing most of my comments. The new results on ImageNets are greatly appreciated too. However, in light of other reviews, I would have hoped that the authors try better training procedures on SWEEN-7 (e.g., MACER even if it means using other ResNets instead of VGG), as otherwise it is difficult to judge whether ensembling really helps. It is unclear why MACER was not used on ImageNet (since all models are ResNets). Overall, the work explores a really important direction of research, but could benefit from further improvements.

---

> ### Author Response · Authors · 2020-11-18
> **Author Response**
>
> We thank the reviewer for positive comments and thoughtful feedback. In the following, we address each point sequentially.
>
> [Regarding SWEEN-7 with MACER]
>
> We found that MACER performs poorly in training models that are not ResNets. For VGG, the optimization may not converge. Since SWEEN-7 contains several such models, i.e., VGG-16 and VGG-19, we only experimented with MACER on SWEEN-3.
>
> We also experimented with adding the ResNet-110 to SWEEN-3 in the MACER scheme. We share the results below in table form. SWEEN-3 + ResNet-110 outperforms both SWEEN-3 and ResNet-110, indicating that SWEEN can consistently improve certified robustness given more candidate models.
>
> | $\\sigma$ | Model                                  | 0\.00 &ensp;| 0\.25 &ensp;| 0\.5 &ensp; | 0\.75 &ensp;| 1\.00 &ensp;| 1\.25 &ensp;| 1\.50&ensp; | 1\.75&ensp; | 2\.00 &ensp;| ACR     &ensp;           |
> |-----------|----------------------------------------|-------|-------|-------|-------|-------|-------|-------|-------|-------|--------------------|
> |           | ResNet\-110                            | 66  &ensp;  | 60 &ensp;   | 53  &ensp;  | 46 &ensp;   | 38 &ensp;   | 29 &ensp;   | 19    &ensp;| 12  &ensp;  | 0  &ensp;   | 0\.726    &ensp;         |
> |     0.5   &ensp;   | UE (SWEEN\-3  ) | 64\.9 | 57\.1 | 49\.7 | 41\.1 | 34\.1 | 26\.2 | 20\.2 | 11\.7 | 0     | 0\.685             |
> |           | SWEEN\-3                               | 64\.7 | 58\.4 | 51\.8 | 43\.9 | 37\.2 | 29\.2 | 22\.8 | 14\.6 | 0     | 0\.727             |
> |           | SWEEN\-3\+ResNet\-110  &emsp;                | 65\.9 | 59\.5 | 52\.1 | 44\.2 | 36\.6 | 30\.4 | 22\.7 | 14\.8 | 0     | $\\textbf\{0\.736\}$ |
>
>
> [Regarding Algorithm 1]
>
> Thanks for this suggestion. We have moved the detailed algorithm to Appendix B.1. As for experiments on ImageNet, the experiments are not finished yet. We will add experiments and replace the adaptive algorithm part once the results are avaliable.
>
> [Regarding ensemble weight training]
>
> Yes. We have tried various losses to train the weights, including cross-entropy, MACER's loss, adversarial loss. The results are quite similar, and we choose the most efficient way. We have changed the formulation to make this point more clear.
>
> [Regarding the $\gamma$-robustness index]
>
> We introduced the notion to make our results more general and clear. Lemma 1 goes beyond the surrogate loss and applies in a broader setting. However, Theorem 2 is restricted to the surrogate loss setting because training is involved.
>
> [Regarding figures]
>
> Sorry for the inconvenience. We added a large version of those figures in Appendix D. We also used different markers and line types to distinguish different models to improve delivery.
>
> We hope our responses can address your concern. We would also be glad to answer any other questions you may have.

---

### Author Response · Authors · 2020-11-18
**Summary of the Initial Revision**

We thank the reviewers for constructive comments.  We have updated a revision of our paper.
Key changes:

1.  Change the formulation of Theorem 2, Theorem 3, Theorem 4, Lemma 10 and remove the condition $l(0, y)=0$.

2. Add comments on certification cost in Section 4.3.

3. Add Appendix B.1 containing Algorithm 1 (moved from the main text)  and addtional comments.

4. Add  Appendix B.2 containing Algorithm 2 which states the general process of training a SWEEN model.

5. Add Appendix D containing full-size figures appeared in the main text.

6. Add FLOPs in Table 3.

7.  Add Appendix C.6 containing the results of SWEEN versus AutoAttack.

8. Fix typos.

In addition, we will update a new revision with experiments on ImageNet when the results are ready.

---

### Author Response · Authors · 2020-11-23
**Results on ImageNet are now available**

We have updated a new revision with experimental results on ImageNet  in the main text. Again, we would like to thank all the reviewers for the insightful comments and constructive feedbacks to help us improve our work. Thank you!

Other changes:

- We  add experiments on ImageNet using models with identical structure (3x ResNet-50) in Appendix C.3

- We redo the experiments in Appendix C.6 with the EoT version of AutoAttack, and the results are almost the same.

- We move the experimental results of Algorithm 1 to Appendix C.4 due to space constraints.

---

### Decision · Program_Chairs · 2021-01-07
**Final Decision**

**Decision:**

Reject

**Comment:**

The paper considers ensambling of smooth classifiers to improve certified robustness. Theoretical results are provided showing that taking ensambles of a large number of models is useful, while experiments show that combining only a small number of models improves performance. On the negative side, the experiments are somewhat inconclusive, as the base models are not state-of-the-art, and the combined results do not achieve state-of-the-art performance. In  this respect, further studies would be necessary to explore the effectiveness of the proposed technique.

In summary, while the topic of the paper is interesting and timely, the proposed ensambling technique is not especially exciting (as it is what one would naturally expect). On the other hand, the problem is reasonably well investigated (e.g., details are worked out well, both theoretical and experimental results are presented), although further experiments are needed (as recommended by the reviewers) to properly assess the potential and limitations of the approach. Accordingly, all reviewers agreed in the discussion that this is a borderline paper. Therefore, unfortunately, it cannot be accepted this time due to the heavy competition at the conference. The authors are encouraged to resubmit a revised version to the next venue, taking into consideration the reviewers' recommendations.